# Agrochemicals increase risk of human schistosomiasis by supporting higher densities of intermediate hosts

Neal T. Halstead [1,14], Christopher M. Hoover[2], Arathi Arakala[3,4], David J. Civitello[5], Giulio A. De Leo [6,7], Manoj Gambhir[3,8], Steve A. Johnson[9], Nicolas Jouanard[10], Kristin A. Loerns[1], Taegan A. McMahon[11], Raphael A. Ndione[10], Karena Nguyen[1], Thomas R. Raffel[12], Justin V. Remais[2], Gilles Riveau[10,13], Susanne H. Sokolow[6,7] & Jason R. Rohr[1]

Schistosomiasis is a snail-borne parasitic disease that ranks among the most important water-based diseases of humans in developing countries. Increased prevalence and spread of human schistosomiasis to non-endemic areas has been consistently linked with water resource management related to agricultural expansion. However, the role of agrochemical pollution in human schistosome transmission remains unexplored, despite strong evidence of agrochemicals increasing snail-borne diseases of wildlife and a projected 2- to 5-fold increase in global agrochemical use by 2050. Using a field mesocosm experiment, we show that environmentally relevant concentrations of fertilizer, a herbicide, and an insecticide, individually and as mixtures, increase densities of schistosome-infected snails by increasing the algae snails eat and decreasing densities of snail predators. Epidemiological models indicate that these agrochemical effects can increase transmission of schistosomes. Identifying agricultural practices or agrochemicals that minimize disease risk will be critical to meeting growing food demands while improving human wellbeing.

[1] Department of Integrative Biology, University of South Florida, Tampa, FL 33620, USA. [2] Division of Environmental Health Sciences, University of California, Berkeley, Berkeley, CA 94720, USA. [3] Department of Epidemiology and Preventive Medicine, Monash University, Melbourne 3800, Australia. [4] Department of Mathematical Sciences, RMIT University, GPO Box 2476, Melbourne 3001, Australia. [5] Department of Biology, Emory University, Atlanta, GA 30322, USA. [6] Hopkins Marine Station, Stanford University, Pacific Grove, CA 93950, USA. [7] Stanford Woods Institute for the Environment, Stanford University, Stanford, CA 94305, USA. [8] IBM Research Australia, Global Services Australia Pvt. Ltd., 60 City Road, Southbank 3006, Australia. [9] Department of Wildlife Ecology and Conservation, University of Florida, Gainesville, FL 32611, USA. [10] Centre de Recherche Biomédicale Espoir pour la Santé, BP 226 Saint-Louis, Senegal. [11] Department of Biology, University of Tampa, Tampa, FL 33606, USA. [12] Department of Biological Sciences, Oakland University, Rochester, MI 48309, USA. [13] CIIL – Institut Pasteur de Lille, 1 Rue du Professeur Calmette, 59019 Lille, France. [14]Present address: Wildlands Conservation, Inc., 15310 Amberly Drive, Suite 250, Tampa, FL 33647, USA. Correspondence and requests for materials should be addressed to N.T.H. (email: neal.halstead@gmail.com)

The global human population is expected to reach approximately 9.7 billion people by 2050[1]. To meet the food demands necessary to support this population, agricultural production is projected to increase 60–70% globally, with fertilizer use increasing 2- to 4-fold and pesticide use 2- to 5-fold relative to levels in 2000[2,3]. Most of the increase in both human population and agrochemical use will occur in developing regions of the world where schistosomiasis is endemic[1–3]. For example, agricultural production is expected to nearly triple in sub-Saharan Africa, the region experiencing the highest population growth rates[3].

Schistosomiasis is caused by trematodes (flatworms) of the genus *Schistosoma* whose transmission relies on freshwater snails that act as an intermediate host. Humans (and various other mammal species) act as the definitive host (the host supporting the adult life stage of the parasite) and are infected when cercariae (the free-swimming life stage of trematodes) released from snails in infested waters penetrate through the skin of the definitive host and mature into adult worms. Global control strategies generally rely on morbidity control through treatment with praziquantel that kills adult worms harbored in human hosts, but drug therapy does not prevent re-infection from future exposure to cercariae. Furthermore, water resources development efforts in many developing countries, particularly the construction of dams and implementation of surface irrigation, have frequently been linked to increased distribution and prevalence of human schistosomiasis[4–8]. Thus, elimination of schistosomiasis has proven difficult throughout most of its geographic extent, with approximately 800 million people living in schistosome-endemic areas (and therefore at risk of infection)[7–9], and at least 218 million people in need of treatment for infection as of 2015[10].

In a trematode-amphibian system that provides a wildlife analog to the schistosome-human system, herbicides and fertilizers increased trematode transmission by stimulating the growth of attached algae (periphyton), the food source for snails (a bottom-up ecological effect)[11,12]. Hence, there is good reason to postulate that agrochemicals might also have important effects on human schistosomiasis. Additionally, insecticides can be deadly to insect and crayfish predators of snails, suggesting that they might increase the number of infected snails by increasing the overall density of snails (a top-down ecological effect)[13], but links between insecticides and wildlife or human trematode infections have not been explored. Here we test the hypothesis that fertilizer, a common herbicide (atrazine), and a common insecticide (chlorpyrifos), individually and as agrochemical mixtures, amplify production of human schistosome cercariae through bottom-up and top-down effects on snail resources and predators, and that this in turn can increase schistosome transmission to humans. We were interested in agrochemical mixtures because they are more commonly detected in nature than individual agrochemicals[13–15].

Here, we use a field mesocosm experiment to demonstrate that environmentally relevant concentrations of agrochemicals (fertilizer, the herbicide atrazine, and the insecticide chlorpyrifos) increase the densities of schistosome-infected snails. These effects occur through both bottom-up effects by increasing the algae snails eat (fertilizer and atrazine) and top-down effects by decreasing densities of snail predators (chlorpyrifos). These effects occur whether agrochemicals are applied individually or as mixtures. In addition, we developed epidemiological models that indicate that these agrochemical effects can increase transmission of schistosomes to humans.

## Results

**Mesocosm experiment**. We created outdoor freshwater pond communities consisting of two snail predators (crayfish:

*Procambarus alleni* and water bug: *Belostoma flumineum*), three snail species (*Biomphalaria glabrata* [native to the Neotropics, introduced to Africa; an intermediate host of *Schistosoma mansoni*], *Bulinus truncatus* [native to Africa, the Middle East, and parts of southern Europe; an intermediate host of *Schistosoma haematobium*], and *Haitia cubensis* [a non-host snail species native to the Caribbean and southeastern United States]), zooplankton, and algae in 60 1200 L mesocosms filled with 800 L of water. *Biomphalaria glabrata* was chosen because laboratory-reared snails were easily available, it is found in both South America and Africa making our results relevant to two continents, and its native range overlaps extensively with that of *H. cubensis* in the Caribbean. We included *H. cubensis* as a non-schistosome host snail species to provide a potential alternative prey source for crayfish predators rather than forcing these predators to only consume schistosome-hosting snails. Agrochemical treatments were applied to the mesocosms in five replicate spatial blocks and consisted of water and solvent (0.0625 mL/L acetone) controls, and atrazine (102 μg/L), chlorpyrifos (64 μg/L), and fertilizer (4400 μg/L N and 440 μg/L P) individually and in all possible combinations (see Methods for details and additional treatments). Globally, atrazine and chlorpyrifos are among the most-used herbicides and insecticides, respectively[16–19], and were applied at their estimated environmental concentrations calculated using US EPA software (see Methods). All three agrochemicals are used commonly in schistosomiasis-endemic regions[17–19]. Mature *S. mansoni* and *S. haematobium* eggs collected from infected Siberian hamsters were added to each mesocosm at three time points to simulate egg introduction from humans in an endemic setting. From each mesocosm, we quantified algal and snail abundance, temperature, light levels, and snail reproduction every other week; *S. mansoni* cercariae shedding rates from *Bi. glabrata* in weeks 8–10; and snail and predator densities as well as snail infection status (for *Bi. glabrata* and *Bu. truncatus*) at the end of the experiment. Additionally, we conducted toxicity tests to evaluate whether there were any ecologically relevant direct effects of the agrochemicals on the egg, miracidium, or cercaria stages of both schistosomes.

A combined factor and path analysis revealed that both top-down and bottom-up effects of the agrochemicals indirectly contributed to increases in infected *Bi. glabrata* densities through increases in overall (infected and uninfected) densities of *Bi. glabrata* (Figs 1a; 2; Supplementary Tables 1, 2). While *Bi. glabrata* was the only snail species for which a sufficient number of infected individuals were alive at the conclusion of the experiment for analysis of the relationship between overall and infected snail densities, treatment effects on the reproductive output and overall densities of all three snail species were significant and in the same direction (Fig. 1a; Supplementary Fig. 1; Supplementary Table 1). Chlorpyrifos reduced densities of crayfish and water bugs (Fig. 1b; Supplementary Fig. 2), which indirectly increased densities of all three snail species by releasing them from predation (Fig. 1c; Supplementary Fig. 1). Both fertilizer and atrazine increased densities of all snail species (when controlling for the effects of predators; Fig. 1d) by increasing algal productivity (Fig. 1e). Fertilizer increased the densities of both suspended and attached algae (Fig. 1a; Supplementary Table 1). Consistent with previous studies[11], atrazine decreased suspended algae and increased the photosynthetic efficiency of attached algae because the reduction in suspended algae increased light availability for the periphyton (linear regression slope between phytoplankton chlorophyll *a* and water column light: coef ± se = −334 ± 166; P = 0.0451). This indirect positive effect of atrazine on attached algae was even greater in the presence of fertilizer (Fig. 1a, Supplementary Table 1).

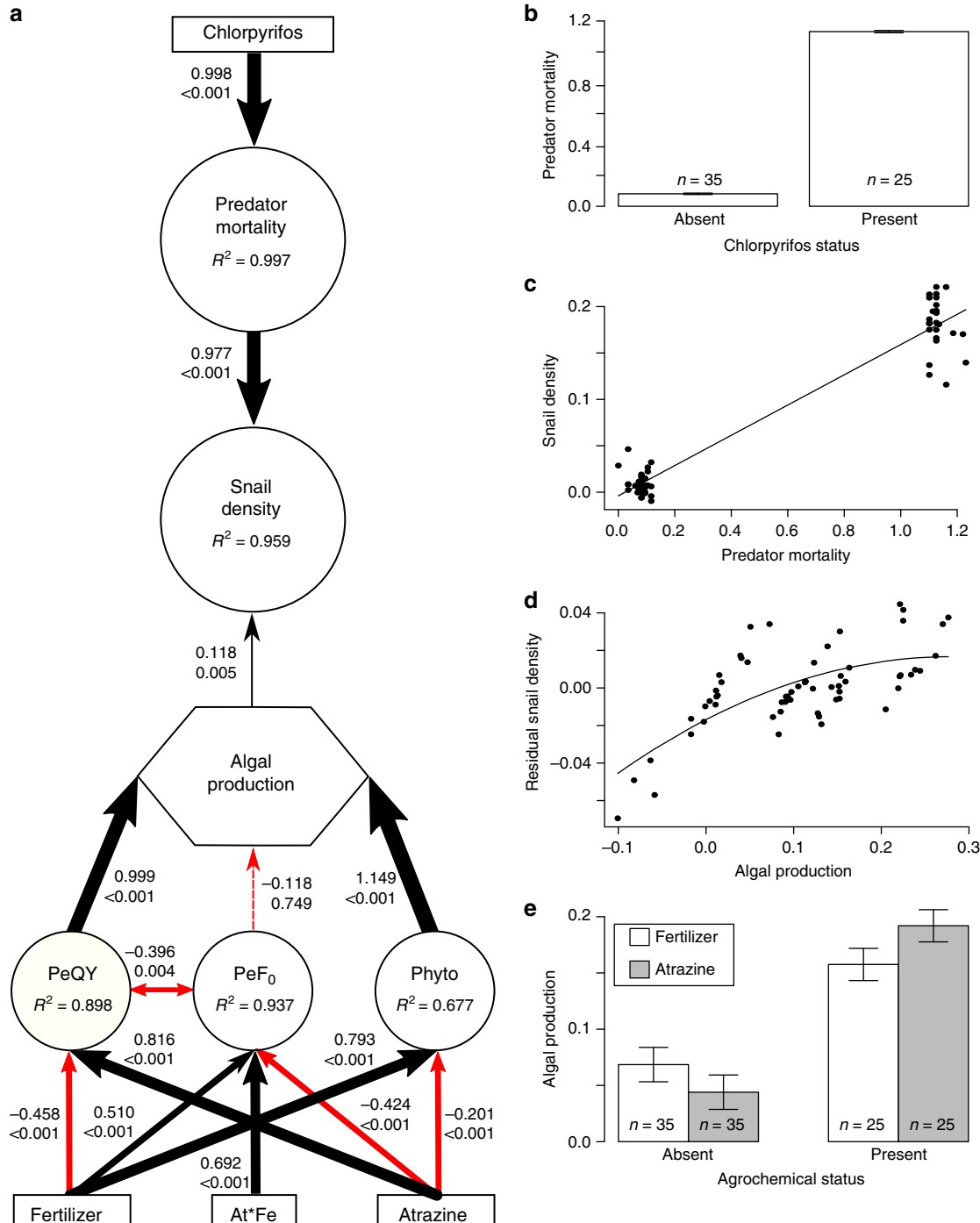

**Fig. 1** Top-down and bottom-up effects of agrochemicals on snail densities. Combined factor and path analysis (**a**), demonstrating top-down effects of chlorpyrifos increasing predator mortality (**b**) and snail density (**c**) and bottom-up effects of atrazine and fertilizer increasing snail density (**d**) through increased algal productivity (**e**). Size of arrows in **a** are scaled to the standardized coefficient (top number next to each arrow), with black and red arrows indicating positive and negative coefficients, respectively. Double-ended arrows exhibit significant covariation accounted for in the structural equation model. *P*-values for paths in the model are reported below each standardized coefficient. Boxes represent exogenous predictor variables, circles represent latent variables, and algal production was measured as a composite variable (hexagon). Indicator variables for latent and composite variables have been omitted from the figure to reduce visual complexity, but are reported in Supplementary Table 1. Importantly, the latent variable snail density represents the densities of all three snail species at multiple life stages (egg, hatchling, and adult), all of which exhibited similar responses across treatments. **e** represents the net main effects of fertilizer and atrazine presence on composite algal productivity. Axes on panels **b**–**e** are derived from latent variable scores for each replicate and thus have no units of measurement; however, raw data are available in the supplemental materials. PeQY = periphyton photosynthetic efficiency; PeF$_0$ = periphyton chlorophyll *a*; Phyto = phytoplankton chlorophyll *a* and photosynthetic efficiency; At*Fe = atrazine x fertilizer interaction term

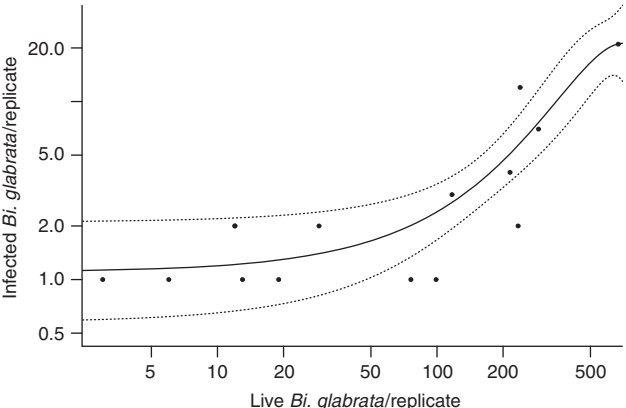

**Fig. 2** Actual number of infected *Biomphalaria glabrata* as a function of live *Bi. glabrata* at the end of the experiment. The response shown is restricted to mesocosm tanks in which infected *Bi. glabrata* were present ($n = 15$ mesocosms), effectively depicting the count portion of the zero-inflated model used to analyze effects on infected *Bi. glabrata* density. However, all 60 replicate mesocosms were used in the analysis (see Methods and Supplementary Table 2 for full model results). Live *Bi. glabrata* density was the only significant predictor of the count of infected *Bi. glabrata* in the model (other than spatial block) and explained 89% of the variation in the density of *S. mansoni*-infected *Bi. glabrata* in replicates in which infected snails were present. The solid line represents the predictions from a generalized linear model with a Poisson distribution including first- and second-degree polynomial terms for live *Bi. glabrata* density as predictors. Dashed lines indicate the 95% confidence band

Top-down regulation of snails by macroarthropod predators, particularly crayfish, was much stronger than bottom-up effects mediated by algal resources in this experiment (Fig. 1). This is consistent with several previous studies that showed that decapod crustaceans are effective biocontrol agents for reducing populations of *Biomphalaria* spp. and *Bulinus* spp.[4,20–24]. Separate field trials using the non-native crayfish *Procambarus clarkii* in Kenya and reintroduction of the West-African native prawn *Macrobrachium vollenhovenii* in Senegal were both successful in reducing *Bulinus* spp. densities and *S. haematobium* infection rates in humans[5,25]. Additionally, reductions in the density of a molluscivorous fish have been linked to increased infection rates and transmission of urinary schistosomiasis in humans[26,27], underscoring the importance of predators in mediating infection dynamics in natural systems[28].

The combined effects of agrochemicals in our mesocosm experiment accounted for 95.9% of the variation in overall snail densities (accounting for all three snail species) in our path model (Fig. 1a; Supplementary Table 1). Overall final densities of *Bi. glabrata* accounted for 89.0% of the variation in the densities of infected *Bi. glabrata* (in replicates with infected snails; Fig. 2) and *Bi. glabrata* density was the only significant predictor of densities of infected *Bi. glabrata* (coef ± se = 0.0056 ± 0.0012; $P < 0.001$; Supplementary Table 2). Together, the indirect effects of agrochemical exposures on snail densities mediated through trophic interactions (top-down and bottom-up effects) and the effect of *Bi. glabrata* density on the density of infected *Bi. glabrata* accounted for 85.4% of the variation in densities of infected *Bi. glabrata*. Importantly, there was no evidence of direct effects of agrochemicals on the number of infected *Bi. glabrata* after controlling for *Bi. glabrata* density (Supplementary Table 2). There also was no evidence of direct effects of agrochemicals on infection prevalence (Supplementary Table 3), cercaria production per snail (Supplementary Table 4), cercarial survival (up to

12 h of exposure; Supplementary Table 5), or schistosome egg viability in toxicity trials (Supplementary Table 6).

**Epidemiological modeling**. To examine the significance of the mesocosm results for human infection, we expanded on classic[29,30] and recent[5] mathematical modeling studies of schistosomiasis transmission by incorporating into models the observed agrochemical effects from our mesocosm experiment, effects from previously published studies examining the same agrochemicals and endpoints as our mesocosm experiment, and parameters fit to previous research on *S. haematobium* transmission to humans in Senegal (see Methods). Our epidemiological model revealed that, in the absence of agrochemical effects and snail predators, the basic reproduction number, $R_0$ (the expected number of mated female worms produced by a single mated female worm in a disease-free setting), was 3.60 (95% CI: 1.32–6.06; Fig. 3a), consistent with previous estimates and the endemic nature of human schistosomiasis in Senegal[5]. The addition of snail predators reduced $R_0$ below 1 (Fig. 3a, b), the minimum threshold for sustained transmission of the disease in the human population, supporting the notion that snail predators can reduce schistosomiasis and protect human health[5]. In contrast, by reducing snail predators, ecologically relevant concentrations of chlorpyrifos increased $R_0$ up to 10-fold relative to controls (Fig. 3a, b, d), suggesting that the removal of snail predators caused by pesticides may lead to a remarkable increase in disease transmission. In the absence of predators or the presence of chlorpyrifos, atrazine and fertilizer further increased $R_0$ by approximately 28% through bottom-up effects (Fig. 3a, c, d).

**Discussion**

To our knowledge, this work represents the first experimental research: 1) to examine the top-down effects of insecticides on trematode transmission; 2) to quantify the top-down and bottom-up effects of agrochemicals on the transmission of human schistosomes; 3) to establish an experimental study system on human schistosomes in outdoor mesocosms; and 4) to link experimental findings on agrochemical effects to human schistosomiasis risk by using parameterized epidemiological models. Given that agrochemical use is expected to rise 2- to 5-fold globally in the next 35 years to meet growing food demands[2], our study has important public health implications in schistosomiasis-endemic regions, as it provides evidence of the potential impact of agrochemicals on the transmission of human schistosomes. Because human population growth rates in schistosome-endemic regions are projected to be much higher than throughout most of the more developed world[1,3], it is likely that the expected 2- to 5-fold mean global increase in agrochemical use will also be proportionally higher in schistosome-endemic regions. Furthermore, environmental conditions (e.g., rainfall intensity and soil characteristics) throughout most schistosome-endemic regions render surface waters highly vulnerable to pesticide runoff[31] and projected increases in agricultural activity in these countries is likely to result in significantly higher probabilities of agrochemical contamination of surface waters, underscoring the need to assess how changes in land use might impact disease transmission[32].

Our results also support recent findings that the presence of generalist predators of snails, such as crayfish (tested here) and the prawn *Macrobrachium vollenhovenii* (native to western Africa) – both of which are omnivores with very similar diets[4,5,20,25,33,34] – can limit or prevent sustained transmission of schistosomiasis (i.e., $R_0 < 1$) by controlling the density of infected snails[5,21,25]. However, consist with previous research demonstrating that the common insecticide chlorpyrifos can induce

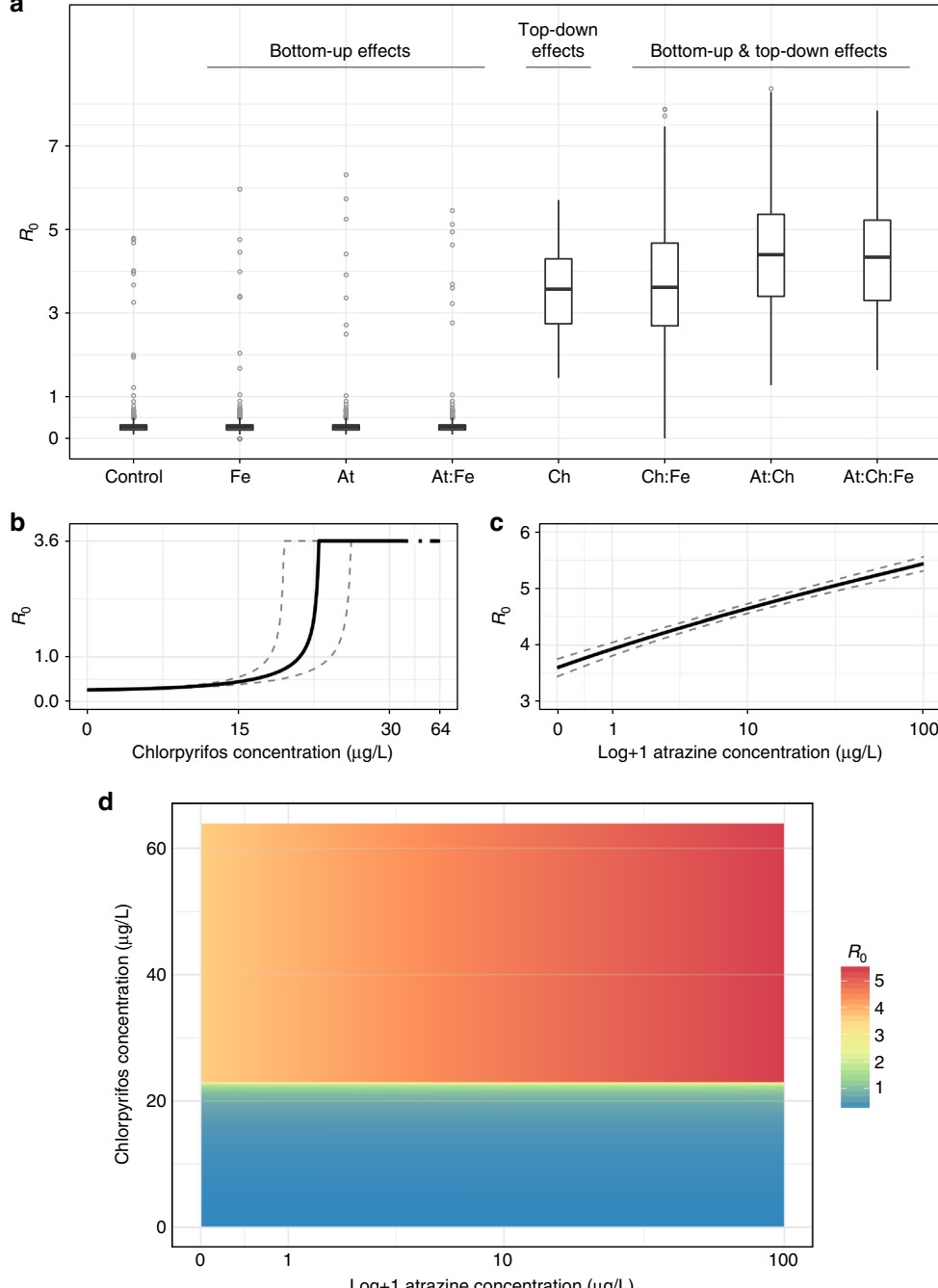

**Fig. 3** Effects of agrochemical treatments on estimates of $R_O$. Results of a mathematical model examining the influence of each agrochemical treatment from the mesocosm experiment on estimates of $R_O$ (**a**), and drawing from other experiments that examine the same agrochemicals and affected pathways to investigate the concentration-dependent $R_O$ of chlorpyrifos (**b**) and atrazine (**c**) as well as their combined influence (**d**). Estimates of $R_O$ were derived from Monte Carlo simulations that incorporate uncertainty associated with both model fitting and agrochemical parameters (see supplementary text). Boxes in **a** represent the median and interquartile range (IQR), whiskers represent values within 1.5*IQR, and outliers (points outside of 1.5*IQR) are plotted as light gray points. When chlorpyrifos is absent, transmission is restricted by top-down regulation of the snail population, causing median $R_O < 1$ in the control and first three treatment groups. In the presence of chlorpyrifos, median $R_O$ estimates are >1, suggesting endemic transmission, and bottom-up effects from atrazine and/or fertilizer act in conjunction with top-down effects to further increase median $R_O$ estimates. Maximum $R_O$ in **b** is achieved when chlorpyrifos concentration is sufficient to eliminate the predator population, as in the mesocosm (64 μg/L). In a predator-free setting, $R_O$ is equivalent to that estimated from the fitted model at baseline, but rises as atrazine concentration (log+1 transformed) increases due to bottom-up stimulation of the snail population. Dashed lines in **b**, **c** represent, respectively, the 95% confidence interval of predator mortality across the indicated range of chlorpyrifos concentration as estimated in ref.[35] and the 95% confidence interval of snail density dependence increases across the indicated range of atrazine concentrations as estimated from ref.[60]. Expected values of $R_O$ driven by mixtures of atrazine acting on snail population dynamics as in ref.[60], and the chlorpyrifos acting on snail predators as in ref.[35] show that agrochemical mixtures have a pronounced influence on transmission intensity (**d**)

considerable mortality in the snail predator population at environmentally relevant concentrations[35], here we demonstrate that the addition of chlorpyrifos results in $R_0$ estimates equivalent to those in predator-free environments that experience endemic transmission (i.e., $R_0 > 1$). Additionally, our results suggest that applications of the common herbicide atrazine and fertilizer can increase the risk of human schistosomiasis in situations where snail predators, such as prawns, exist at densities too low to effectively regulate snail populations (e.g., where dams have been constructed or in surface irrigation canals[4,5,7,8]). To ensure that schistosomiasis-endemic regions can address their current and pending human malnutrition crises without increasing schistosomiasis, it will be important to implement farming practices that minimize agrochemical runoff, continue advances in the sustainable use of snail biological control agents[5], and identify agrochemicals that can increase food production without increasing snail densities.

## Methods

**Mesocosm experiment.** We established outdoor freshwater ponds in 1200 L mesocosms filled with 800 L of water at a facility ~20 miles southeast of Tampa, FL, USA. The tanks were filled with tap water on 4 June 2010 and allowed to age for 48 h before being seeded with algae (periphyton and phytoplankton) and zooplankton collected from local ponds on 6 June 2010. Algal and zooplankton communities were allowed to establish over a four-week period and 40 L of water was mixed between tanks weekly to attempt to homogenize initial communities before application of agrochemical treatments. Sediment (1 L play sand and 1 L organic topsoil (The Scotts Company, Marysville, OH, USA)) was added to each tank on 1 July 2010 (Supplementary Fig. 3). Immediately before application of agrochemical treatments on 8 July 2010, snails (27 *Biomphalaria glabrata* (NMRI strain), 11 *Bulinus truncatus* (Egyptian strain), and 30 *Haitia cubensis*) and snail predators (3 crayfish (*Procambarus alleni*), and 7 giant water bugs (*Belostoma flumineum*)) were added to each tank. Initial snail and predator densities were chosen within ecologically relevant limits[36–38] and determined by availability from either the NIAID Schistosome Resource Center (for laboratory passaged strains of *Bi. glabrata* and *Bu. truncatus*) or local availability for *H. cubensis*, *P. alleni*, and *B. flumineum* from wetlands in the Tampa, FL, area. More specifically, densities of *P. alleni* in South Florida exhibit a high degree of spatiotemporal variability, and can exceed 10 crayfish m-2, but are typically significantly lower[36]. We chose a density on the lower end of observed natural densities (0.6 crayfish m-2) because 1) our initial stocking densities for snail species were also relatively low, 2) crayfish presence/absence (rather than density) determined the variation in snail survival and recruitment in previous mesocosm studies[13,39], and 3) we wanted to limit our impact on local source populations of crayfish (given that we had 60 mesocosms to supply with predators). Finally, our initial crayfish densities were nearly identical to the maximum natural prawn densities reported in the literature (~0.6 prawns m-2)[38], which is also the estimated density of prawns necessary to extirpate local snail populations and well above the threshold density of prawns needed to reduce/eliminate schistosomiasis locally (~0.3 prawns m-2)[5], but significantly lower than aquaculture densities that can exceed well over 10 prawns m-2[40,41]. The mesocosm experiment was approved under USF Institutional Biosafety Committee Study number 0971.

Sixty tanks were randomly assigned (using a random number generator) to one of 12 treatments in 5 replicated spatial blocks. Water and solvent (0.0625 mL/L acetone) controls were used to ensure that any observed effects in agrochemical treatments were not due to the presence of solvent. The herbicide atrazine and the insecticide chlorpyrifos were dissolved in acetone and applied at their respective estimated peak environmental concentrations (EEC: atrazine = 102 μg/L; chlorpyrifos = 64 μg/L), determined using USEPA's GENEEC (v2.0, USEPA, Washington, D.C.) software, manufacturers' label application recommendations, and the physicochemical properties of each pesticide (Supplementary Table 7). Target concentrations of fertilizer (N: 4,400 μg/L, P: 440 μg/L) were based on ponds identified as highly productive in a field survey conducted by Chase[42]. Hence, the agrochemical concentrations selected for this study are environmentally relevant and well within the ranges observed in the field. Fertilizer was applied as a mixture of sodium nitrate and sodium phosphate dissolved in acetone. Each chemical was applied individually at its EEC, at 2x the EEC, and in all pairwise combinations. The 2x EEC treatments were included as an additional reference to account for pairwise mixtures having approximately twice the amount of chemicals added. Technical-grade pesticides were used for all treatments (purity > 98%; Chemservice, West Chester, PA, USA) and actual concentrations of pesticides applied to the replicates were confirmed using ELISA test kits (Abraxis, LLC, Warminster, PA, USA) in the Rohr lab. ELISA assays were calibrated by using standards of known concentration for each pesticide, or calculated from established cross-reactivity to the chemical used to determine the standard curve. For any nominal concentrations below the limit of detection for the kit, we confirmed the concentration of the stock solution used for serial dilutions.

*Schistosoma mansoni* (NMRI strain) and *S. haematobium* (Egyptian strain) eggs were collected from infected Siberian hamsters and added to mesocosms at three separate occasions during the experiment. Eggs were added on multiple occasions after application of agrochemical treatments to better simulate the relatively constant input of schistosome eggs into waterbodies as opposed to more infrequent pulses of agrochemical runoff into surface waters. Snails and infected hamsters were provided by the NIAID Schistosomiasis Resource Center. Five infected hamsters were euthanatized on 27 July 2010, 4 August 2010, and 12 August 2010), and *S. haematobium* eggs were collected from the intestines. Eggs were isolated from tissue using a handheld immersion blender and collected on a 45 μm USA standard test sieve (Fisher Scientific, Pittsburgh, PA, USA). Mature eggs were stored in a 1.4% NaCl solution to inhibit hatching in a 50 mL centrifuge tube. Eggs were suspended repeatedly using a vortex mixer and sixty-five 3 mL aliquots were prepared for each schistosome species and added to the tanks within two hours of collection. An additional three aliquots were preserved to quantify the total number of eggs added to each tank. Egg viability was quantified by placing subsamples of the remaining mature eggs in artificial spring water[43] and observing the proportion of hatched miracidia within 1 h. The mean number of *S. mansoni* eggs in each aliquot was 981.1 (±46.5 SEM) eggs, with a mean viability of 29.4% (±4.6% SEM), which resulted in an estimated 289 *S. mansoni* miracidia added to each tank at each weekly addition. The mean number of *S. haematobium* eggs in each aliquot was 2276.7 (±107.5 SEM) eggs, with a mean viability of 8.5% (±1.9% SEM), which resulted in an estimated 193 *S. haematobium* miracidia added to each tank at each weekly addition. Collection of schistosome eggs from infected hamsters was approved by animal care and use committee protocols T 3829 and R 3517 at the University of South Florida.

Strict biosafety protocols were established and approved by USF Biosafety (IBC #1334) to minimize the risk of infection to researchers and escape of snails from the mesocosms. Researchers working at the mesocosm facility wore personal protective equipment, including shoulder-length PVC gloves (#7451, Galeton, Foxborough, MA, USA), when removing or replacing items in tanks. In addition, all researchers working on the mesocosm experiment had blood drawn before and several months after conducting the experiment. Blood samples were sent to the Centers for Disease Control and Prevention to test for schistosome infections. Tanks had an inward-projecting outer rim along the top edge, were only filled halfway, and were covered with heavily weighted shade cloth to prevent snail escape or entry of any large organisms. The mesocosm facility was surrounded by two layers of silt fence with molluscicide (1.0% iron phosphate; Natria®, Bayer Advanced, Research Triangle Park, NC, USA) applied between the fences at the recommended rate of 1 pound per 1000 square feet every two weeks during the experiment. Tanks were a minimum of 200 m from the nearest waterbody and the entire facility was surrounded by chain link and barbed wire fencing. At the end of the experiment, each tank was over-treated with pool shock (71.8% trichloro-s-triazinetrione, Recreational Water Products, Buford, GA, USA; applied at 0.15 g/L) to kill all of the snails and schistosomes before the tanks were emptied and the snails were removed and preserved.

Periphyton measurements were recorded from 100 cm² clay tiles suspended vertically 15 cm from the bottom of each tank (approximately 20 cm below the water's surface), facing south along the northern wall of each tank. Five clay tiles were added to each tank when they were initially filled with water. Algal samples were collected immediately prior to agrochemical addition (Week 0) and at 1, 2, 4, 8, and 12 weeks post-application. Phytoplankton and periphyton chlorophyll *a* and photosynthetic efficiency (measured as $F_0$ and QY, respectively), were measured from samples stored in darkness for 1 h, using a handheld fluorometer (Z985 Cuvette AquaPen, Qubit Systems Inc., Kingston, Ontario, Canada). Temperature and light levels were quantified on the same dates as algal sampling by suspending a data logger (HOBO Pendant UA-002–64, Onset Computer Corporation, Bourne, MA, USA) 20 cm below the water surface for 30 minutes in each replicate within a spatial block near midday. Loggers were rinsed in tap water after the 30-minute data collection period for each block before being transferred to the next spatial block to avoid cross contamination of agrochemicals. Midday temperatures 20 cm below the water surface were 32.55 ± 0.10 °C on 9 July 2010, 30.49 ± 0.08 °C on 15 July 2010, 31.47 ± 0.10 °C on 22 July 2010, and 31.29 ± 0.07 °C on 5 August 2010 (all mean ± se).

Snail reproductive effort and density was estimated using two 15 × 30 cm pieces of Plexiglass placed in each tank; one suspended vertically 10 cm from the bottom of each tank and one resting horizontally along the tank bottom (see Supplementary Fig. 3). Snail egg masses, juveniles, and adults were quantified from each sampler at weeks 1, 2, 4, 8, and 12. Visual searches for dead *P. alleni* and *B. flumineum* occurred 24 and 48 h after agrochemical addition, and upon each snail sampling session. Ten weeks after agrochemical addition, pool shock was added to each tank as described above to kill any infective schistosome cercariae and tanks were subsequently drained through a kick net (800/900 μm, 425-K11, Wildlife Supply Company, Yulee, FL, USA) to collect remaining organisms. All snails and macroinvertebrates were collected, fixed in formalin for one week, and subsequently preserved in 70% ethanol. Snail infection status was determined by cracking each snail's shell and inspecting the hepatopancreas and gonads under a dissecting microscope. Researchers were not blind to treatments when collecting or analyzing data.

There was no effect of solvent on any response variables, so solvent and water controls were pooled and treated as a single control treatment. Likewise, there was no effect of concentration on any of the observed response variables, so 1 × and 2 × EEC single pesticide treatments were combined for analysis. Photosynthetic efficiency was logit-transformed prior to analysis. All other response variables were natural log+1 transformed.

Structural equation modeling was used to explore combined causal pathways of pesticide mixtures using the lavaan package[44] in R statistical software[45]. Because a sample size of 60 tanks restricted the number of causal pathways we could infer, we first constructed a latent variable for predator mortality (P. alleni and B. flumineum mortality at 24 h and the end of the experiment) and a second model consisting of latent variables for phytoplankton production ($F_0$ and QY from weeks 1–8), periphyton chlorophyll a ($F_0$ from weeks 1–4), and periphyton photosynthetic efficiency (QY from weeks 1–4). Model comparison using AICc was performed to select the best latent variable model from alternative configurations of indicator variables (i.e., algal parameters as separate or combined latent variables without a composite variable). The scores for each latent variable model were then extracted using the predict function and used for construction of a structural equation model that included snail response variables (number of egg masses on snail samplers from weeks 1–4, the number of snail hatchlings on snail samplers from weeks 4–8, and the number of live snails collected at the end of the experiment for each species, including the non-host H. cubensis) as indicators of a latent variable of overall snail density.

The relationship between infected Bi. glabrata density and density of live Bi. glabrata at the end of the experiment was analyzed using generalized linear models in the pscl package[46,47] in R statistical software[45]. This relationship is only presented for Bi. glabrata because there were too few infected Bu. truncatus alive at the end of the experiment to perform the same analysis. Final Bi. glabrata density in addition to fixed main effects of agrochemicals, their interactions, and block were used as predictors of the count of infected Bi. glabrata in each tank, with a zero-inflated Poisson distribution (Supplementary Table 2). We also tested for direct effects of agrochemicals on infection prevalence using a beta binomial error distribution of infected vs uninfected Bi. glabrata in each tank with fixed effects of agrochemicals, their interactions, and block as predictors (Supplementary Table 3). Model selection indicated that the beta binomial error distribution was a better fit to the prevalence data than a binomial distribution (ΔAICc = 8.3). Analysis of prevalence data was performed using the glmmADMB package[48,49] in R[45]. Light availability was tested as a response of phytoplankton chlorophyll a and fixed effects of block in week 2, using the glmmADMB package in R.

The observed algal dynamics are consistent with previous research[11]. Fertilizer increased phytoplankton density (chlorophyll a) and photosynthetic efficiency, and increased periphyton density, but photosynthetic efficiency of periphyton was reduced in fertilizer treatments. However, phytoplankton density was a negative predictor of light availability in the water column (coef ± se = −334 ± 166; P = 0.0451), and decreased light availability is therefore likely to reduce the photosynthetic efficiency of periphyton. Conversely, atrazine decreased phytoplankton chlorophyll a and photosynthetic efficiency. Thus, although atrazine negatively impacted periphyton chlorophyll a, photosynthetic efficiency of periphyton increased in the presence of atrazine because more light was available for photosynthesis, and a positive interaction between the joint presence of atrazine and fertilizer in mixtures increased periphyton density substantially (Fig. 1, Supplementary Table 1). The lack of complex refugia for snails in the mesocosms may have artificially decreased the apparent strength of bottom-up effects on snail densities relative to top-down regulation by predators. We explored the potential for a submerged macrophyte, Hydrilla verticillata, to provide refugia for snails in a separate mesocosm experiment, and found no evidence that H. verticillata provided effective refugia from omnivorous crayfish (see Supplementary Methods, Supplementary Fig. 4, Supplementary Table 8). However, because crayfish readily consume both living and decaying plant matter, non-consumable refugia might provide a stronger mediating effect on the relative strengths of top-down versus bottom-up regulation of snail populations.

**Cercaria production experiment**. To test for indirect effects of agrochemical exposure on cercarial production (through potential effects of agrochemicals on resource composition, quality, and/or abundance), fifteen freshwater mesocosms were established at the same time and using the same methods as noted above for the main mesocosm experiment, with the exception that no snails or snail predators were added to these tanks. Instead, algal and zooplankton dynamics were allowed to respond to agrochemical treatments in the absence of periphyton herbivores. This mesocosm experiment was also approved under USF Institutional Biosafety Committee Study number 0971.

Tanks were randomly assigned to one of 5 treatments in 3 replicated spatial blocks. Atrazine, chlorpyrifos, and fertilizer were applied at their respective EECs as described previously. Solvent controls were used to account for the presence of solvent used to deliver agrochemicals in solution. In addition, a treatment combining atrazine and fertilizer at their respective target concentrations was included to test for a potential interaction between these two agrochemicals. Technical-grade pesticides were used for all treatments and actual concentrations of chemicals applied to the replicates were confirmed as described above for the main mesocosm experiment.

Forty infected Bi. glabrata (NMRI strain) were obtained from the NIAID Schistosomiasis Resource Center and added to the first block of tanks four weeks post-miracidia exposure (snails exposed 28 July 2010, added to first block 24 August 2010). Eight snails were added to each of the five replicate tanks in one block and left in the tanks for three days before calculating cercaria production rates. On the third morning after adding snails to each tank, snails were removed from each tank and placed individually in 250 mL specimen containers filled with 100 mL of ASW for 1 h. After 1 h, snails were removed from each container and five drops of Lugol's iodine were added to preserve and stain cercariae. Snails were haphazardly assigned to replicate tanks in the next spatial block and left for three days before repeating the cercaria production trials. This process continued until the infected snails were rotated through each block a total of two times. Cercariae were counted in the laboratory under a dissecting microscope.

We tested for main effects of agrochemical treatment on the total number of cercariae shed per hour with a negative binomial generalized linear model, using the glmmADMB package in R[48,49]. We used fixed main effects for each chemical and days since miracidia exposure and included random effects of tank nested in block nested by trial number (first or second trial) as predictors of the cercaria shedding rate. In addition, we tested for an interaction between atrazine and fertilizer in the absence of chlorpyrifos. Cercaria production rates per snail increased with increasing time since miracidial exposure (coef ± se = 0.138 ± 0.035; P < 0.001; Supplementary Table 4), but there were no effects of agrochemical exposure (all P > 0.35; Supplementary Table 4).

**Direct effects of agrochemical exposure on schistosome cercariae**. To test for direct effects of agrochemical exposure on the cercariae of S. mansoni, six replicates of four agrochemical treatments (atrazine, chlorpyrifos, fertilizer and a solvent control each at their EEC as described above for the mesocosm experiment) were randomly assigned to the wells of a 24-well tissue culture plate (Falcon® # 353047, Corning Incorporated, Corning, NY, USA) containing freshly collected Schistosoma mansoni (NMRI strain) cercariae in 400 μL of COMBO[50] (8.65 ± 0.64 cercariae/well). One-hundred μL of stock solution of each agrochemical was added to randomly assigned wells at the beginning of each trial to reach the target EEC for each treatment at a total volume of 500 μL. Survival of cercariae was assessed at 2, 4, 8, 12, and 24 h after agrochemical addition using a separate 24-well plate for each time point. At the given end point for each trial, the number of dead cercariae was determined by adding 15 μL of trypan blue, a selective stain that is taken up only by dead cercariae[51], to each well. Following staining with trypan blue, 20 μL Lugol's iodine was added to each well to kill and stain all cercariae in the well.

Cercarial survival at each time point was tested using a binomial generalized linear model in the glmmADMB package in R[45,48,49]. Fixed main effects of each agrochemical and time were used as predictors of the proportion of dead cercariae in each well. There was a significant negative effect of time since agrochemical exposure on cercarial survival (coef ± se = −0.264 ± 0.017; P < 0.001), but not of any agrochemicals (all P ≥ 0.10). When analyzing cercarial survival at each time point independently, no main effects of agrochemicals were evident within 12 h of exposure to agrochemicals (all P ≥ 0.10; Supplementary Table 5). Chlorpyrifos and fertilizer each had significant negative effects on cercarial survival at 24 h post-exposure (Supplementary Table 5). However, because infectivity of S. mansoni cercariae declines rapidly and is very low beyond 8–15 h[52], any treatment effects of agrochemicals after 12 h are less ecologically relevant than earlier time points.

**Schistosome egg viability experiment**. To test for direct effects of agrochemical exposure on the egg viability of S. mansoni and S. haematobium, we conducted standard toxicity trials on the hatching rates of schistosome eggs exposed to agrochemicals. Eggs were collected from the tissues of two S. mansoni-infected Swiss-Webster mice and two S. haematobium-infected Siberian hamsters, on 1 Sep 2011, 6 September 2011, and 8 September 2011. See Methods for the mesocosm experiment for detailed methods on egg collection. Eggs were stored in 1.4% NaCl to prevent hatching before beginning egg viability trials on each day. For each species on each date, twelve agrochemical treatments (described above) were randomly applied to wells filled with 1.0 mL ASW in two spatial blocks in a 24-well tissue culture plate (Falcon® # 353047, Corning Incorporated, Corning, NY, USA). After applying agrochemicals to each well, approximately 20 eggs of either S. mansoni or S. haematobium were added to each well. The number of miracidia in each well was counted after 1 h. Lugol's iodine was then added to each well to stain and count the unhatched eggs in each well. One plate trial was performed on each date for each species, for a total of six replicate trials per species. Egg viability was tested with a beta binomial generalized linear mixed-effects model, using the glmmADMB package in R[45,48,49]. Fixed main effects of and interactions between agrochemicals and random effects of block nested within plate were used as predictors of hatching success. No main effects of agrochemicals or interactions between agrochemicals on schistosome egg viability were evident (all P > 0.05; Supplementary Table 6).

**Modeling experiments**. A model expanding on previous[5] and classic[29,30] work was used to investigate agrochemical effects—of atrazine, chlorpyrifos, and fertilizer—on human schistosomiasis transmission intensity. The model includes snail population dynamics of Bulinus truncatus, the intermediate host of Schistosoma haematobium, subject to logistic population growth and the influence of predation.

We focus on *S. haematobium* for our modeling because it is the predominant schistosome species found in the village in Senegal where the epidemiological data used to parameterize our model were collected. The population dynamics of generalist predators ($P$) are included, subject to an agrochemical-sensitive mortality rate, $\mu_{P,q}$, that reflects chlorpyrifos toxicity to the predator population as estimated by the mesocosm experiments and previous work[35]. Because crayfish and prawns are generalist predators and will switch to other resources when snail densities are low, the model assumes that predator population dynamics are independent of snail density. Also included is a parameter representing agrochemical enhancement of the snail carrying capacity, $\varphi_{N,q}$, which models the snail population response to bottom-up effects caused by algal stimulation by atrazine and fertilizer as estimated in the mesocosm experiments and other experiments examining the same agrochemicals and outcomes. Additional model state variables represent susceptible, exposed and infected snails ($S$, $E$, and $I$, respectively) and the mean worm burden in the human population ($W$). The number of mated female worms, $M$, is estimated assuming a 1:1 sex ratio and mating function, $\gamma(W, k)$, as in ref.[53]. The per capita snail predation rate by predators, modeled as a Holling type III functional response as in ref.[54], $\psi$, and the total snail population, $N$, are shown separately for clarity. Parameter values, definitions and reference literature are listed in Supplementary Table 9.

$$\frac{dS}{dt} = f_N \left( 1 - \frac{N}{\varphi_N * \varphi_{N,q}} \right)(S + E) - \mu_N S - P\psi S^n - \beta MS \quad (1)$$

$$\frac{dE}{dt} = \beta MS - \mu_N E - P\psi E^n - \sigma E \quad (2)$$

$$\frac{dI}{dt} = \sigma E - (\mu_N + \mu_I)I - P\psi I^n \quad (3)$$

$$\frac{dW}{dt} = \lambda I - (\mu_H + \mu_W)W \quad (4)$$

$$\frac{dP}{dt} = f_P \left( 1 - \frac{P}{\varphi_P} \right)(P) - (\mu_P + \mu_{P,q})P \quad (5)$$

$$M = 0.5WH\gamma(W, k) \quad (6)$$

$$\psi = \frac{\alpha}{1 + \alpha T_h N^n} \quad (7)$$

$$N = S + E + I \quad (8)$$

**Derivation of $R_0$.** Using the next generation matrix method[55], we calculate $R_0$ for this system as:

$$R_0 = \sqrt{\frac{T_1 T_2}{T_3}}$$

Where:

$$T_1 = 0.5\beta HN^*$$

$$T_2 = \lambda\sigma$$

$$T_3 = (\mu_W + \mu_H)\left(\mu_N + \frac{P^*\psi^*}{3} + \sigma\right)\left(\mu_N + \frac{P^*\psi^*}{3} + \mu_I\right)$$

$$\psi^* = \frac{\alpha N^{*n-1}}{(1 + \alpha T_h N^{*n})}$$

and $N^*$ and $P^*$ represent disease-free equilibrium values of the snail and predator populations, respectively, derived by setting equations (1) and (5) equal to 0 such that:

$$P^* = \left(1 - f_P^{-1}\left(\mu_P + \mu_{Pq}\right)\right)\varphi_P$$

and $N^*$ is estimated by solving for $N^*$ in the polynomial:

$$0 = f_N\left(1 - \left(\varphi_N \varphi_{Nq}\right)^{-1}N^*\right) - \mu_N - P^*\psi^*$$

The model expressed as equations (1) through (8) was fit to previously published worm burden data from a baseline and follow-up survey of *Schistosoma haematobium* infection in a rural community upstream of the Diama Dam in Senegal[5]. The human-to-snail transmission parameter, $\beta$, and two values of the snail-to-human transmission parameter, $\lambda_{lo}$ and $\lambda_{hi}$, were fit to the seasonal reinfection data using maximum likelihood estimation in R with the *optim* function[45], with all other parameters held to values shown in Supplementary Table 9, and agrochemical and predation effects turned off (i.e., $\varphi_{Nq} = 1$ and $P = 0$). Estimates of uncertainty associated with model fitting were generated by exploring the three-dimensional parameter space around the best-fit values of $\beta$, $\lambda_{lo}$, and $\lambda_{hi}$ (shown in Supplementary Table 9) by varying each plus or minus 90% of its best-fit value. Assuming the negative log likelihood profile follows a chi-square distribution with three degrees of freedom, all parameter triplets that have negative log likelihood within 7.815 (95% CI, two-sided chi-square critical value) of the negative log likelihood produced by the best-fit values are within the 95% confidence interval. These parameter triplets were used in Monte Carlo simulations described further below to generate estimates of $R_0$. When estimating steady-state transmission indices such as $R_0$ a time-weighted average of $\lambda_{lo}$ and $\lambda_{hi}$ was used.

Point estimates of the baseline daily predator mortality rate, $\mu_P$, and the chlorpyrifos-enhanced predator mortality rate, $\mu_{P,q}$, were derived directly from 24 h mortality endpoints in the mesocosm experiment by treating all *Procambarus alleni* in chlorpyrifos tanks (75 total) as a treatment cohort and all *Procambarus alleni* in chlorpyrifos-free tanks (105 total) as a control group (Supplementary Table 9). A parametric distribution of the predator mortality rate was obtained by fitting beta distributions to 5000 bootstrapped samples of daily predator mortality in each of the 25 mesocosm tanks with and 35 mesocosm tanks without chlorpyrifos added (Supplementary Table 10).

To investigate the influence of a broader range of chlorpyrifos concentrations on estimates of $R_0$, a probit model of predator mortality was derived spanning the range of chlorpyrifos concentrations (0–64 μg/L) tested in ref.[35], conservatively assuming 99% mortality in the highest tested concentration groups instead of 100% to account for potential resistance in a small number of predators. Daily, per capita predator mortality rates were derived across the range of tested concentrations and used in the $R_0$ expression to generate Fig. 3b–d and Supplementary Fig. 5.

Because crayfish are generalists, we modeled their predation of snails using a Holling type III functional response (eqn 7) in which the per capita predation rate is sigmoidal due to prey switching at low snail densities and restriction by the handling time ($T_h$) at high snail densities[54,56,57]. Though not directly interpretable, the exponent, $n$, of the type III functional response is often assumed to be 2 for invertebrate predators[56,58]. However we also tested a range of values from 1–4 for the exponent, $n$, and found little qualitative difference in results when $n > 1$. When $n = 1$, the functional response reduces to a Holling type II in which the predation rate increases rapidly at low prey density and asymptotes at high prey densities where predation is restricted by the handling time. In our model, this leads to predation-induced extirpation of the snail population and $R_0 = 0$, a result we would not expect in real-world transmission settings in which we expect prey switching by the predator population as well as refuge-seeking by snails to diminish predatory activity at decreasing snail densities.

Bottom-up effects in the mesocosm were introduced through model parameter $\varphi_{N,q}$, a scalar that represents a proportional change in the baseline snail density-dependence parameter, $\varphi_N$. To quantify the effect of atrazine and fertilizer, alone and in combination, on this parameter while controlling for the strong influence of predators on snail population dynamics, we calculated the mean proportional increase in final *Bu. truncatus* density in the mesocosm tanks when fertilizer and atrazine were added in combination with chlorpyrifos (Supplementary Table 10). We found no previous studies in which chlorpyrifos had a significant direct effect on snail population dynamics nor on algal dynamics at the concentrations tested in the mesocosm. The distribution of the density dependence scalar, $\varphi_{N,q}$, was obtained by fitting normal distributions to 5000 bootstrapped samples of the parameter estimate derived from individual tanks within each treatment group (Supplementary Table 10). To further investigate the influence of atrazine at concentrations below the maximum expected environmental concentration—as was tested in the mesocosm—we derived another atrazine-dependent scalar of the carrying capacity based on the results of ref.[59]. Briefly: the scalar was calculated according to the proportional increase in the peak growth rate at the tested atrazine concentrations over the observed peak growth rate of the control group, as discussed in refs. [60].

To produce estimates of $R_0$ that incorporate uncertainty associated with both model fitting and agrochemical parameterization, we ran 1000 Monte Carlo simulations for each agrochemical treatment and the control group; drawing randomly from the agrochemical parameter probability distributions described above and from the range of best-fit transmission parameters (Supplementary Table 10). The probability of sampling particular transmission parameter triplets was weighted by a normalized index of their likelihood so that triplets that better fit the model were more likely to be included in the simulation.

**Code availability.** All scripts containing the code used for data analysis in R are available from the corresponding author upon request.

**Data availability**. The experimental data that support the findings of this study are publicly available in figshare with the identifier https://doi.org/10.6084/m9.figshare.5797389.

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

## Acknowledgements

We thank G. Agemy, Z. Babwani, J. Cook, E. Cooper, A. Earls, A. Gilbert, J. Jones, M. Kepner, S. Kilgore, B. Mathew, M. McGarrity, J. Rivera, A. Rodríguez, A. Tapilyai, and C. Towne for assistance with collecting data in the field and laboratory. Snails and infected hamsters were provided by the NIAID Schistosomiasis Resource Center. This research was supported by grants from the National Science Foundation (EF-1241889), National Institutes of Health (R01GM109499, R01TW010286), US Department of Agriculture (NRI 2006-01370, 2009-35102-0543), and US Environmental Protection Agency (CAREER 83518801) to J.R.R., and National Institutes of Health (K01AI091864) and the National Science Foundation (EAR-1646708, EAR-1360330) to J.V.R.; a University of Florida Research Innovation Award to J.R.R. and S.A.J.; an Oakland University Research Excellence Fund award to T.R.R; and Dana and Delo Faculty Development Grants from the University of Tampa (T.A.M.). S.H.S. and G.A.D.L were supported by National Science Foundation (CNH grant # 1414102), the Bill and Melinda Gates Foundation, National Institutes of Health (R01GM109499, R01TW010286-01), Stanford GDP SEED (grant # 1183573-100-GDPAO), the SNAP-NCEAS working group "Ecological levers for health: Advancing a priority agenda for Disease Ecology and Planetary Health in the 21st century" and the NIMBioS-supported working group on the Optimal Control of Environmentally Transmitted Disease.

## Author contributions

J.R.R. conceived the experiment. N.T.H. and J.R.R. designed the experiment. S.A.J. provided field and laboratory facilities. N.T.H., T.A.M., K.A.L., K.N., T.R.R. and J.R.R. conducted the experiment. K.A.L. determined infection status of snails. N.T.H. and D.J.C. conducted the statistical analyses. N.J., G.R. and R.A.N. collected field data for parameterization of the epidemiological modeling. C.M.H., A.A., M.G. and J.V.R. conducted the mathematical modeling and risk analysis. G.A.D.L. and S.H.S. participated to the calibration of the model and to the analysis of model results. N.T.H., C.M.H. and J.R.R. wrote the manuscript and all authors contributed to its editing.

## Additional information

**Competing interests:** The authors declare no competing financial interests.

