## [Peer Review File · Nature Communications]

Reviewers' comments:

Reviewer #1 (Remarks to the Author):

Using an impressive field mesocosm experimental set-up, the authors of this manuscript wish to test the hypothesis that a common insecticide, herbicide and fertilizer (individually and as agrochemical mixtures) amplify production of human schistosome cercariae and that this in turn can increase schistosome transmission to humans. To do so, the authors created 60 outdoor freshwater pond units consisting of two snail predators (the crayfish *Procambarus alleni* and the water bug, *Belostoma flumineum*), two intermediate host snail species (*Biomphalaria glabrata* and *Bulinus truncatus*, intermediate hosts of *Schistosoma mansoni* and *S. haematobium*, respectively), one non-host snail species (*Haitia cubensis*), some zooplankton, and algae.

The results suggest that mainly top-down (reduced predator densities) and to some extent bottom-up effects (more available resources through increased algae production) of the agrochemicals indirectly contributed to increases in infected snail densities through changes in overall snail densities.

In general, the ms is very well-written. The set-up of the experiment seems well thought-through and carefully designed. The epidemiological models build to some extent on previous developed models developed by the authors, and appear to be appropriate given the data at hand. Hence, I will focus in the remainder of this review on the level of novelty of the results and the potential wider implications of these.

Novelty and relevance: The authors claim that their study has important public health implications in schistosomiasis endemic regions, as they here provide first evidence of the potential impact of agrochemicals on the transmission of human schistosomes. And certainly, the results point to some potential very relevant impacts of increased use of agrochemicals on snail-borne parasites in Africa in particular.

However, I would like to contest the claims of the authors on a few points:

First, as the authors themselves point out, although is the first time this kind of experiment is carried out with human schistosomes, it is not the first time that a study has shown that agrochemicals can have this effect on snail-borne disease transmission, through increased snail densities: The Rohr et al study, 2008, conducted mesocosm experiments with amphibian snail-borne trematodes that showed that, atrazine increased snail abundances up to 4 times, due to more food resources for the snails. However, in the latter study, the causal mechanisms established in the mesocosm experiments, was nicely supported by the analysis of data from field surveys, giving it a much stronger base to make broader conclusions from. Naturally, the fact that we are dealing with a human final host in the case of schistosomiasis, makes it more challenging. But using a combination of data from field surveys and manipulative studies as conducted here, would not be impossible, and would, if a correlation between schistosomiasis and the use of these agrochemicals can be found, substantially corroborate the claims made by the authors in this ms. Especially since human schistosomiasis is so massively multifactorial in etiology, with snail densities playing only a partial role in the transmission dynamics, as the authors no doubt are very well aware.

Another reason why it would be important to test if the results from the experiments hold under field conditions, is a few weaknesses in the experiment itself:

1: I am surprised that there is not a stronger bottom-up effect relative to the top-down effect. Organic loading for instance has often been shown to be a very important factor at many transmission sites in Africa (because of the increased food resources for the snails). The authors do point to the fact that there are no refugia in the tanks, and that could be part of the explanation. Instead they find a very

strong top-down effect through decreased predation. However, for this effect to be meaningful in a natural setting there would of course have to be crayfish, crabs or prawns predated substantially on the snail populations at the transmission sites to begin with, let alone be present in significant numbers. Again, field surveys could help clarify to what extent this is really the case..

2: Two out of the three snail species in the experiment are not native to Africa: *Bi. glabrata* is a South American species, (not the main host in Africa, which is *Biomphalaria pfeifferi*). While *Bi. glabrata* often is used in experiments (because it is easy to keep in a lab-setting), *Bi. pfeifferi* does have different requirements. Also, the non-host snail species is not African, but a local species to the mesocosm-experiment. While this no doubt is a matter of practicality, it should be made clear, since the authors refer to African schistosomiasis studies and conditions through-out the ms.

3: The effect of agrochemicals on the abundance of other snails in the community (competitors and/or predators), and the composition of the snail community as a whole, is likely to be substantial. Here, only one other snail species is added to the experiment (and it is not a species found in endemic transmission sites). Also, the effect on this snail species in the experiment is not discussed in the ms.

4: Finally, there are no records of what temperatures the experiments were conducted at? As temperature is one of the most important determinants of both snail and parasite reproduction and development, it would be relevant for the readers to know if the temperatures in the field mesocosm were comparable to typical endemic schistosomiasis transmission sites in Africa. Also, it would be interesting to know what, if any, the effect of varying temperature would be on the outcome of the experiment.

Reviewer #2 (Remarks to the Author):

Summary

The manuscript describes an extensive mesocosm experiment examining the role of mixtures of pesticides and fertilizer on a semi-realistic aquatic community relevant for understanding human pathogen transmission. The authors test commonly applied pesticides at ecologically relevant doses on two human parasites, *Schistosoma mansoni* and *S. haematobium*. Furthermore, the aquatic communities consist of multiple trophic levels, including algal food sources for snails and two types of snail predators. This allowed the researchers to distinguish the influence of top-down and bottom-up effects by using an appropriate and rigorous combined factors and path analysis statistical approach. Key findings include significant effects on the infected snail intermediate hosts from both directions. Overall, top-down effects of predation had a stronger effect than bottom-up effects on primary producers. This was further supported by smaller scale laboratory experiments investigating direct toxicity to parasite life stages. Building on the results of the mesocosm and laboratory experiments, the researchers used a mathematical modeling approach building on previous literature and incorporated the results of the agrochemical treatments to make predictions for human health. The modeling analysis revealed a significant effect of the loss of snail predators on the basic reproduction number, R_0 of human schistosomiasis due to the mortality caused by agrochemicals. Taken together, the results of the mesocosm experiment, laboratory experiments on toxicity, and the mathematical modeling extend similar results in wildlife pathogen systems and support the idea that human population growth and increased agrochemical use in schistosome endemic areas will lead to important human health implications.

General Comments

The main claims of this paper are that agrochemicals have indirect effects mediated by the aquatic food web on the density of infected intermediate hosts of a human pathogen. The novelty of the paper

stems from the approach of using mesocosm experiments with the actual human pathogens as opposed to previous research on trematodes of wildlife that have similar life cycles. Because of the type of experimental design incorporating community level interactions, this study has a broad appeal for disease ecologists, epidemiologists, parasitologists, and general ecologists interested in food web interactions. However, I do think the results are confirmatory of the previous studies in wildlife disease systems. The direct application to a human disease system is important, but I do not think it will move our thinking about the field in a new direction, rather reinforce the ideas that agrochemicals can have indirect, negative effects on host-pathogen interactions.

In contrast to the main strengths of the manuscript there were a few areas where the paper could be strengthened, specifically in explanation of the experimental design and interpretation. First, what was the rationale/justification for including the non-host snail? Why was this included and how did it factor into the analysis? It is unclear why this snail was included and how would it represent the interactions the mesocosm is intended to simulate? Furthermore, the manuscript is vague and unclear about what "snail densities" are throughout. Does this mean only the potential host snails, all snails or either *Biomphalaria glabrata* or *Bulinus truncatus*? It seems like the authors are lumping all the snails together and it makes it very difficult to separate out the true effects and interpretation.

A related issue, was how the manuscript lumped schistosomes together at some points, focused on *Bi. glabrata* and *S. mansoni* for some experiments and generally ignored the effects on *Bulinus* and *S. haematobium* and any interactions between the species. For example, both snails and parasites are included in the main mesocosm experiment and in the laboratory egg viability experiment, but only cercariae from *S. mansoni* were tested in the toxicity mesocosms (data in Extended data table 4). Specifically, in extended data table 1 – there are statistically significant effects from the structural equation model of the other species of snails *Bu. truncatus* and *H. cubensis* that are not discussed in the paper. In extended data table 2 only *S. mansoni* infected *Bi. glabrata* are analyzed. Extended data table 3 is unclear about whether these results combined *S. mansoni* and *S. haematobium*. Finally, the epidemiological model is of *S. haematobium*, but the rest of the results are primarily *S. mansoni* (Line 416-417). The manuscript seems to jump between the two species and use them almost interchangeably. While these taxa are often discussed together in other literature, in this manuscript it was confusing and potentially misleading. I would encourage the authors to clarify and be more transparent about their results for both species separately and any assumptions they are making about the responses of one taxa, whether snail or parasite, for the other.

Another area for improvement is the interpretation and context for the strong negative effect of the pesticide chlorpyrifos on the snail predators. First, very few predators were actually added to each mesocosm, particularly only 3 crayfish, so that mortality of even one individual would have a huge effect (>33% or >14% mortality with only one death in crayfish and water bugs respectively) and in a ~10 week long experiment reproduction of predators may not have been possible to replace mortality (unlike the rapid snail population growth). If all the predators died that would be expected to have dramatic effects. Furthermore, the effects of predation were inflated by the lack of structure in the mesocosms for prey refugia and lack of alternative prey for predators. Follow up experiments with snails and predators, or a more thorough incorporation of the literature on predation among these species would be helpful in interpreting whether the large effects of predation are representative or exaggerated by weaknesses of the experimental design. Additional experiments would be easier to support because biosafety precautions associated with the parasite exposure may not be necessary unless predation rate differs between infected and uninfected snails. In general, I also noticed limited context from the previous literature on the role of predators on infected hosts. Instead, the literature used to support the strong top down effects was based on predation on parasite free-living infective stages, which was not what the study was investigating. Predation on infected hosts is a different mechanism than predation on parasites with potentially other effects on the aquatic communities. The

authors should support their results with relevant examples from the extensive literature on the role of predation on infected hosts.

Unfortunately, I believe these issues, as well as other minor issues provided in the specific comments below, prevent the manuscript from being acceptable in the present form. However, I do believe that there is also significant merit in the research and if the authors can make substantial improvements in these areas that a resubmission of the manuscript should be considered.

Specific Comments

The manuscript was an appropriate length and written concisely, but there are several important issues that could be clarified specifically related to the experimental design and interpretation of the mesocosm experiment. There were also a couple instances where I thought the authors were overselling their claims and made some suggestions to highlight the importance of the findings without going too far. Finally, there were some instances where more methodological detail is needed. One area is the parasite egg amounts and timing to the mesocosms and how this interacts with snail density. In the methods and extended data section the authors did provide adequate information so that others could replicate mathematical modeling as well as detailed information about the biosafety precautions to the researchers and the native environment to prevent infection or release of organisms. However, for interpretation it would be helpful to include information about the densities of organisms at the end of the experiment, specifically mortality data on the predators, which is currently only incorporated into variables for modeling. Fig1 shows many key relationships, but are actually taken from the path analysis rather than actual experimental values and makes it difficult to compare to previous literature because of the unit-less, experiment specific measures.

Abstract:

Line 23: Explain what is meant by intensification. Does this mean prevalence, intensity, or some other measure?

Line 25: Explain increases. Does this mean increase in area irrigated or permanency? The term is unclear.

Line 29: Where is the agrochemical use predicted to increase by 2050? Is this globally or in areas where schistosomes could be present. Please be more specific.

Line 30: I think this statement extends the results too far. While the combined experiments and modeling demonstrate potential mechanisms that could lead to increasing schistosomiasis generally the applicability is far from global and many other conditions were not considered (for example, prey switching by predators, refugia for prey) that would potentially influence the results across both small and large geographic scales. I recommend revising to better reflect the scope of the results.

Line 37: Insert "agricultural" after "identifying".

Main text:

Line 48: delete "cycle"

Line 49: Please define the terms "definitive" and "cercariae". Readers may be unfamiliar with the meaning of these terms.

Line 54: Are there more up to date values of infection risk? 2003 is quite old.

Line 70: What is the rationale for including both *Biomphalaria glabrata* and *Bulinus truncatus*?

Furthermore, what is the rationale for including the *Haitia cubensis*? How does this represent the natural environment the mesocosms should be modeling? See the general comments above for additional considerations.

Line 79-80 & Line 248-259: How many eggs were added to each mesocosm for each parasite species? How did the numbers of eggs added on different days compare? Both this section and the methods are vague and do not report the actual numbers added. While it is stated that this would simulate egg introduction from humans in an endemic setting there is no information given about how the density of egg input represents natural communities.

Furthermore, how did the timing of egg addition influence the results? For example, snails were in the tanks reproducing uninfected for about 3 weeks before parasite addition. I presume that agrochemical induced changes could result in higher parasite densities where parasites were added. However, the parasites castrate the snail, which could influence reproduction, so it is unclear how high infection rates of the snails would then influence snail density. How does the timing of the mesocosm additions represent a natural system?

Line 83: Insert "laboratory" before "toxicity"

Lines 104-106: Here and elsewhere (Line 351-355), the authors draw from studies on the diversity of predators on parasite life stages to support their results of predation having an important effect on transmission rates. However, the types and effects of the predation are not the same. Predation on parasite life stages could reduce the standing crop of infective stages, but the predation on the intermediate hosts eliminates further parasite production by killing the host. These are two distinct pathways that are not necessarily interchangeable. It is unclear if these predators could consume the parasites themselves in addition to their effects on the snail population. Furthermore, there is extensive literature on the role of predation on disease dynamics and that could be better incorporated here as well as elsewhere (Line 351-355) in the manuscript.

Line 109: Please see the general comments above for additional discussion. The term "snail densities" is used throughout, but it is unclear if the authors are talking about all three snails, just the schistosome host snails (*Bi. glabrata* and *Bu. truncatus*), or one or the other. These results are difficult to interpret without more information about the study design. Throughout the results the authors need to be explicit about what the responses are and by which taxa, particularly when the results tables are only included in the extended tables making it difficult for the reader to access them directly within the manuscript. In the methods when the analyses are discussed and in the discussion of the results the distinction and effects of the different snails should be made clearly. Results for the non-host snail are presented in Extended table 1 and are reported as being statistically significant, but this is never mentioned in the manuscript.

Line 118: Remove "indicating that the density of infected snails should reliably indicate schistosome exposure risk to humans". The study did not directly test this and it is an overstatement of the experimental results.

Line 141-144: How similar are these species? I would like to see further evidence to support this claim.

Figures:

Fig. 1A: I think the figure is misleading by not including the pathways from fertilizer and atrazine. The lack of numerical values by the lines makes it look as if these pathways don't have significant effects when they do. The representation of the figure over-emphasizes the role of predation, while the

bottom-up factors are pushed back to extended data figure 1. While I can understand the authors' desire to keep the figure simple and interpretable, it would be better to have both parts of the figure included.

Figure 1E: I would revise the x-axis label to be Agrochemical status or treatment. It was confusing at first to see absent as a value for agrochemical presence.

Furthermore, as stated in the specific comments above, I think it would be helpful to present figures of the actual data rather than the output of the structural equation model. The unit-less measures are not transparent and prevent the data from being compared with the literature.

Fig. 3D. Missing the letter – also the color scheme did not reproduce well in B&W. I would suggest changing the color scheme so that the trends are still apparent when reproduced in other formats.

Methods:

Line 218: What was the amount of water?

Line 222: What is the rationale/justification for the different snail densities and particularly for the predators?

Line 326: Does snail density refer only to the potential host snails? Clarify.

Line 486: The authors acknowledge some lack of realism in the mesocosm experiment, but could a follow up study be done to examine prey-switching and refuge-seeking by the snails with the predators to help examine the strength of these effects and the potential influence on the results?

Author contributions

Not all authors indicated on the title page are included in the author contributions (Missing DJC).

Extended Data

Extended Data Figure 1. Please see comments for Figure 1.

Extended Data Table 1. Please explain the significant effects of the non-host snail as well as clarify the interactions of the two schistosome hosts with each other.

Extended Data Table 2. This table only represents *Bi. glabrata*. What about *Bu. truncatus*?

Extended Data Table 3. Does this table represent the total infected snail density with *S. mansoni*, *S. haematobium* or just one or the other?

Reviewer #3 (Remarks to the Author):

Title: Agrochemical pollution increases risk of human exposure to 2 schistosome parasites

3 Authors: Neal T. Halstead^{1*†}, Christopher M. Hoover², Arathi Arakala³, David J. Civitello⁴,

⁴ Giulio A. De Leo⁵, Manoj Gambhir³, Steve A. Johnson⁶, Kristin A. Loerns¹, Taegan A.

⁵ McMahon⁷, Thomas R. Raffel⁸, Justin V. Remais², Susanne H. Sokolow⁵, Jason R. Rohr¹

An interesting and potentially very important topic – proposing that the current trend towards increased agrochemical use can, through increasing snail intermediate host densities (predominantly through reducing snail predators), increase the numbers of infected and uninfected snails, and thereby

potentially increase the global burden of schistosomiasis. In particular, using a combination of experimental and modelling simulations, the authors demonstrate that: i) commonly-used agrochemicals can induce mortality in the snail predator population at environmentally relevant concentrations, resulting in R_0 estimates equivalent to those in predator-free environments; ii) snail and infected snail densities may also increase due to feeding on increased algae present.

The first question raised is whether agrochemical increased usage globally/equally distributed – as schistosomiasis, a disease of animals as well as the humans presented here, is a highly focal disease and impacts the poorest of the poor populations – ones which may not afford agrochemicals on their subsistence farms next to small, often transient, waterbodies? However, the authors do support this with cited evidence in support of mass increases in agrochemical usage within sub-Saharan Africa, but may still be worth considering – in particular in relation to the e.g. large lake v small pool waterborne risks of schistosomiasis in Africa.

The second, is that how relevant is this to standard snail-predator-free localities, outside those biological-control prawn/crayfish trials successfully ongoing in Senegal and Kenya? However, I do appreciate if snail densities do also increase per se within areas of high agrochemical pollution, relative to pollutant-free sites, then this should be the case.

Some clarification within each of the methodology subsections would be of use to the reader – in regards to what specific hypothesis/aim each component is addressing.

Indeed in general it would have been of interest to know how viable the cercariae were being shed from such infected snails – where one could propose that, whilst potentially increasing the snail densities, could such agrochemicals present in the water be serving as cercariacides and thereby reducing the risk to subsequent human infections? I appreciate no such impact on the eggs was observed.

To ensure that schistosomiasis-endemic regions can address their
156 current and pending human malnutrition crises without increasing schistosomiasis, it will be
157 important to implement farming practices that minimize agrochemical runoff

Valid points, wherever feasible. Hence I do think, with some minor clarifications, this article will be of good general interest.

A few specific comments or queries the authors may like to address:

Greater than 10% of the global population is at risk of schistosomiasis,

Interesting way of presenting this, a focal/NTD, but impressive if accurate.

Some potential text proof-reading might help – such as '[The] global human population...To meet the food demand[s] necessary....

Humans act as the
49 definitive host

Humans act as definitive hosts for human schistosomiasis – as of course there are plenty of animal schistosomes also.

when cercariae released from snails in infested waters burrow

50 through skin a

Cercariae do penetrate but I'm not sure 'burrow' is the appropriate term.

with an

54 estimated 779 million people at risk of infection as of 2003.

55

There are much more recent estimates of numbers at risk since 2003.

common insecticide (chlorpyrifos), a common herbicide (atrazine),

I presume they are two commonly used within schistosome-endemic zones/developing countries. Ok, I see they get to this by lines 76-78 (p4)

(*Biomphalaria glabrata* [the intermediate host of *Schistosoma mansoni*], *Bulinus truncatus* [the 70 intermediate host of *Schistosoma haematobium*],

[an] intermediate host for each – there are many alternatives.

Also any details on these snail lines? Presumably laboratory passaged?

Figure 2. Actual number of infected *Biomphalaria glabrata* as a function of live *Bi. glabrata* at 178 the end of the experiment

Are these real data? How can the number of infected snails go up over time – are the eggs/miracidia not just presented at the beginning of the experiment?

Or was it at three time points: Five infected hamsters were euthanized on 27 July 2010, 4 August 2010, and 12 August 2010),
And matched per tank?

This has to be made clearer within the text.

Methods

Immediately before application

222 of agrochemical treatments on 8 July 2010, snails (27 *Biomphalaria glabrata* (NMRI strain), 11 *Bulinus truncatus* (Egyptian strain), and 30 *Haitia cubensis*) and snail predators (3 crayfish 224 (*Procambarus alleni*), and 7 giant water bugs (*Belostoma flumineum*)

Specify why these somewhat random and non-matched numbers of snails were selected?

Mature eggs were

254 stored in a 1.4% NaCl solution to inhibit hatching in a 50 mL centrifuge tube. Eggs were 255 suspended repeatedly using a vortex mixer and sixty-five 3mL aliquots were prepared for each 256 schistosome species and added to the tanks within two hours of collection.

Ok, but would it not have been better/more standardized to hatch the miradia and give each replicate tank a precise set number?

Snail infection status was determined by

304 cracking each snail's shell and inspecting the hepatopancreas and gonads under a dissecting
305 microscope.

Ok, but simply 'shedding' the snails for viable infections at the end of study may have been useful.

Cercaria Production Experiment
358 Experimental design

Are the authors now testing for cercarial viability with agrochemical present? Within each sub-section of the material and methods, some explanation of the specific hypothesis/aims being tested would be very useful.

Indeed it would be very interesting to know if, whilst potentially increasing the snail densities, could such agrochemicals present in the water be serving as cercariacides and thereby reducing the risk to human infections.

Ah nice, I see they are testing this in terms of the eggs at least, although no impact on egg viability was observed.

After applying agrochemicals to each well, approximately 20 eggs of either *S. mansoni* or *S. haematobium* were added to each well.

The modelling used appears appropriate.

Reviewers' comments:

Reviewer #1 (Remarks to the Author):

Using an impressive field mesocosm experimental set-up, the authors of this manuscript wish to test the hypothesis that a common insecticide, herbicide and fertilizer (individually and as agrochemical mixtures) amplify production of human schistosome cercariae and that this in turn can increase schistosome transmission to humans. To do so, the authors created 60 outdoor freshwater pond units consisting of two snail predators (the crayfish *Procambarus alleni* and the water bug, *Belostoma flumineum*), two intermediate host snail species (*Biomphalaria glabrata* and *Bulinus truncatus*, intermediate hosts of *Schistosoma mansoni* and *S. haematobium*, respectively), one non-host snail species (*Haitia cubensis*), some zooplankton, and algae. The results suggest that mainly top-down (reduced predator densities) and to some extent bottom-up effects (more available resources through increased algae production) of the agrochemicals indirectly contributed to increases in infected snail densities through changes in overall snail densities.

In general, the ms is very well-written. The set-up of the experiment seems well thought-through and carefully designed. The epidemiological models build to some extent on previous developed models developed by the authors, and appear to be appropriate given the data at hand. Hence, I will focus in the remainder of this review on the level of novelty of the results and the potential wider implications of these.

Thank you for the kind words. They are greatly appreciated.

Novelty and relevance: The authors claim that their study has important public health implications in schistosomiasis endemic regions, as they here provide first evidence of the potential impact of agrochemicals on the transmission of human schistosomes. And certainly, the results point to some potential very relevant impacts of increased use of agrochemicals on snail-borne parasites in Africa in particular.

However, I would like to contest the claims of the authors on a few points:

First, as the authors themselves point out, although is the first time this kind of experiment is carried out with human schistosomes, it is not the first time that a study has shown that agrochemicals can have this effect on snail-borne disease transmission, through increased snail densities: The Rohr et al study, 2008, conducted mesocosm experiments with amphibian snail-borne trematodes that showed that, atrazine increased snail abundances up to 4 times, due to more food resources for the snails. However, in the latter study, the causal mechanisms established in the mesocosm experiments, was nicely supported by the analysis of data from field surveys, giving it a much stronger base to make broader conclusions from. Naturally, the fact that we are dealing with a human final host in the case of schistosomiasis, makes it more challenging. But using a combination of data from field surveys and manipulative studies as conducted here, would not be impossible, and would, if a correlation between schistosomiasis and the use of these agrochemicals can be found, substantially corroborate the claims made by the authors in this ms. Especially since human

schistosomiasis is so massively multifactorial in etiology, with snail densities playing only a partial role in the transmission dynamics, as the authors no doubt are very well aware.

The reviewer is correct that this is not the first time that a study has shown that agrochemicals can have such effects on snail-borne disease transmission, and the Rohr et al. 2008 study did indeed provide substantial evidence for the bottom-up effects of atrazine specifically on wildlife trematodes. However, that study did not provide any experimental manipulations of fertilizer, nor did it provide any evidence that insecticides affected trematode infections through top-down effects. We have revised the ms to highlight the novelty of these aspects of the present work. Moreover, the Rohr et al. 2008 study was conducted on a completely different set of snails and trematodes, most of which do not infect humans.

We also note that research conducted on model systems is frequently published in top general science journals (e.g., Nature Communications, Nature, Science); take for example breakthrough infection or drug experiments in model mouse or rat systems. These studies are often followed by additional papers in these same journals based on the results of clinical trials. Both types of studies make important contributions, and the confirmatory work in human studies builds on the critical foundation that controlled experimentation provides (and, of course, overcomes the lingering doubts over whether results from well-established “model” mammalian species can be reliably used to predict patterns in human systems). Studies on model systems provide the proof of concept, and observational epidemiological studies and then clinical trials verify that the results in humans are similar to those observed in the model organisms. From this important set of research traditions, we acknowledge that human schistosomiasis is highly multifactorial in etiology, and we take care in the revised ms to ensure that researchers do not assume that the effects of a single herbicide on a snail-amphibian trematode system should match the effects of this chemical on a snail-human trematode system.

Another source of novelty that we now highlight in the ms is that, to the best of our knowledge, there has never been a manipulative mesocosm experiment on human schistosomes. Mesocosms are enormously powerful. They are manipulative and thus can investigate causality, yet they also reflect the complexity of the natural world better than highly controlled laboratory experiments because they include natural food resources, multiple trophic levels, and natural fluctuations in temperature, sunlight, pH, dissolved oxygen, etc. To provide just a glimpse into the tremendous undertaking mesocosm research on this human pathogen is: the experimental work required approvals from USDA APHIS, the Schistosomiasis Resource Center, the Department of Environmental Protection, and the University of South Florida’s Biosafety, IACUC, and Environmental Health and Safety Offices. In addition, it required testing all the staff regularly by sending blood samples to the CDC. To properly contain the snails and pathogens, we placed tanks inside of tanks, only filled the tanks half full, placed pool shock in the outer tanks, covered the tanks with shade cloth so nothing can get in or out, surrounded the entire tank array with two layers of construction sediment fencing, placed molluscicide between the sediment fencing, and surrounded the entire facility with barb-wired chain link fence, as detailed in lines 306-323. The massive mobilization of research infrastructure, compliance and safety precautions needed to successfully conduct these experiments is a key component of the novelty of the work, and we believe that documentation of the results from such a major research feat is an important contribution to science.

Finally, another novel aspect of our study is the linking of the mesocosm work to actual human risk through parameterized epidemiological human schistosome transmission models. This is something that no previous studies on wildlife trematodes have done. We believe this greatly enhances the rigor and scope of this work and further supports our conclusions that these chemicals can increase human risk.

In summary, we believe that our work is novel because, to our knowledge, this work represents the first experiment 1) to examine the top-down effects of insecticides on trematode transmission, 2) to quantify the top-down and bottom-up effects of agrochemicals on the transmission of human schistosomes, 3) to study human schistosomes in outdoor mesocosms, and 4) to link its findings on agricultural development to human schistosomiasis risk by using parameterized epidemiological transmission models. We have added this sentence to the ms (lines 158-163) to better highlight its novelty. Moreover, given the extreme human population growth expected in tropics in the coming years, the interactions between food production and schistosomiasis risk that we explore in our ms have enormous public health and policy implications for the developing world.

The reviewer is correct that the Rohr et al. 2008 paper did indeed provide field evidence to support the mesocosm results and we agree that similar data here would be ideal. However, this is far from a trivial undertaking given that both agrochemical concentrations in waterbodies and human urine and fecal samples need to be collected and analyzed from numerous replicate villages in the developing world. We have received funds and completed all the necessary compliance steps to conduct this very work in Senegal, and we initiated the first field survey this month. But, it will be at least a full year before we have one season of data and two or more years to have sufficient data to assess the correlations the reviewer suggests. Noting the policy relevance of these issues, and because we believe that there is value in providing the evidence we have collected to-date with limited delays, we chose to carry out our risk analysis using mathematical models, evaluating the likelihood that our mesocosm findings could affect human infection risk. We completely understand the reviewers' and the editor's concern, but we chose not to delay the publication of this research for another 2 or more years; we hope, in the mean time, that this work stimulates more work by other groups on this important topic.

Another reason why it would be important to test if the results from the experiments hold under field conditions, is a few weaknesses in the experiment itself:

1: I am surprised that there is not a stronger bottom-up effect relative to the top-down effect. Organic loading for instance has often been shown to be a very important factor at many transmission sites in Africa (because of the increased food resources for the snails). The authors do point to the fact that there are no refugia in the tanks, and that could be part of the explanation. Instead they find a very strong top-down effect through decreased predation. However, for this effect to be meaningful in a natural setting there would of course have to be crayfish, crabs or prawns predated substantially on the snail populations at the transmission sites to begin with, let alone be present in significant numbers. Again, field surveys could help clarify to what extent this is really is the case.

Prawns have historically been extremely widespread in Senegal where the epidemiological work in the ms was conducted. We have clarified this and added citations to highlight this (lines 121, 172, 555-556).

To address the reviewers concerns regarding a lack of physical structure in our mesocosms, we conducted a mesocosm experiment on schistosomes and their predators across a gradient of macrophyte densities. In this study, we show that low densities of the same predators used in the experiments currently in the ms completely eliminated adult recruitment of snails in both the presence and absence of macrophytes that provided structure and refugia in these mesocosms. The results of these experiments have now been added to the extended data of the ms (lines 494-580, Extended Data Figure 2, Extended Data Table 8).

2: Two out of the three snail species in the experiment are not native to Africa: *Bi. glabrata* is a South American species, (not the main host in Africa, which is *Biomphalaria pfeifferi*). While *Bi. glabrata* often is used in experiments (because it is easy to keep in a lab-setting), *Bi. pfeifferi* does have different requirements. Also, the non-host snail species is not African, but a local species to the mesocosm-experiment. While this no doubt is a matter of practicality, it should be made clear, since the authors refer to African schistosomiasis studies and conditions throughout the ms.

3: The effect of agrochemicals on the abundance of other snails in the community (competitors and/or predators), and the composition of the snail community as a whole, is likely to be substantial. Here, only one other snail species is added to the experiment (and it is not a species found in endemic transmission sites). Also, the effect on this snail species in the experiment is not discussed in the ms.

Great points. We address points 2 and 3 together because they are related. We chose to add an alternative non-schistosome harboring host because we know that prey-switching and alternative food sources can greatly impact predation pressures that predators can exhibit. We were worried that readers might assume that we were over-estimating predation rates if we forced the predators to only consume snails that can harbor human schistosomes (i.e., by only providing these species). Getting the full suite of potential food sources of snail predators in western Africa was not logistically possible. We felt that adding another snail species was important to offer an alternative prey item and improve the robustness of our findings.

*We have revised the ms to make it clearer where the snail species are native (lines 75-78), why we added the local snail species, and the possibility that the results could depend on the availability of alternative prey items (lines 82-84). We also added to the ms that the effects of the treatments on the local snail species did not significantly differ from the effects on *Bi. glabrata* or *Bu. truncatus*, emphasizing the robustness of our findings (lines 103-106). The upshot of including *Bi. glabrata* is that our results might be relevant to schistosomiasis risk across two continents, Africa and South America, which we also now highlight in the ms (lines 79-81).*

4: Finally, there are no records of what temperatures the experiments were conducted at? As temperature is one of the most important determinants of both snail and parasite reproduction and development, it would be relevant for the readers to know if the temperatures in the field mesocosm were comparable to typical endemic schistosomiasis transmission sites in Africa. Also, it would be interesting to know what, if any, the effect of varying temperature would be on the outcome of the experiment.

Great point. This was an omission on our part. We used Hobo loggers that simultaneously measured both light and temperature. Hence, we have temperature measurements on each tank throughout the experiment. We now provide weekly estimates of the mean and variance of temperature for each treatment in our experiment in the supplement (lines 337-339).

Reviewer #2 (Remarks to the Author):

Summary

The manuscript describes an extensive mesocosm experiment examining the role of mixtures of pesticides and fertilizer on a semi-realistic aquatic community relevant for understanding human pathogen transmission. The authors test commonly applied pesticides at ecologically relevant doses on two human parasites, *Schistosoma mansoni* and *S. haematobium*. Furthermore, the aquatic communities consist of multiple trophic levels, including algal food sources for snails and two types of snail predators. This allowed the researchers to distinguish the influence of top-down and bottom-up effects by using an appropriate and rigorous combined factors and path analysis statistical approach. Key findings include significant effects on the infected snail intermediate hosts from both directions. Overall, top-down effects of predation had a stronger effect than bottom-up effects on primary producers. This was further supported by smaller scale laboratory experiments investigating direct toxicity to parasite life stages. Building on the results of the mesocosm and laboratory experiments, the researchers used a mathematical modeling approach building on previous literature and incorporated the results of the agrochemical treatments to make predictions for human health. The modeling analysis revealed a significant effect of the loss of snail predators on the basic reproduction number, R_0 of human schistosomiasis due to the mortality caused by agrochemicals. Taken together, the results of the mesocosm experiment, laboratory experiments on toxicity, and the mathematical modeling extend similar results in wildlife pathogen systems and support the idea that human population growth and increased agrochemical use in schistosome endemic areas will lead to important human health implications.

General Comments

The main claims of this paper are that agrochemicals have indirect effects mediated by the aquatic food web on the density of infected intermediate hosts of a human pathogen. The novelty of the paper stems from the approach of using mesocosm experiments with the actual human pathogens as opposed to previous research on trematodes of wildlife that have similar life cycles. Because of the type of experimental design incorporating community level interactions, this study has a broad appeal for disease ecologists, epidemiologists, parasitologists, and general ecologists interested in food web interactions. However, I do think the results are confirmatory of the previous studies in wildlife disease systems. The direct application to a human disease system is important, but I do not think it will move our thinking about the field in a new direction, rather reinforce the ideas that agrochemicals can have indirect, negative effects on host-pathogen interactions.

Thank you for the kind words regarding our ms and highlighting its novelty. We encourage the reviewer to see the comments above about the top-down effects and modeling being completely unique aspects of this work that have never been explored in any wildlife studies.

In contrast to the main strengths of the manuscript there were a few areas where the paper could be strengthened, specifically in explanation of the experimental design and interpretation. First, what was the rationale/justification for including the non-host snail? Why was this included and how did it factor into the analysis? It is unclear why this snail was included and how would it represent the interactions the mesocosm is intended to simulate?

Great question. This should have been made clearer in the ms. We chose to add an alternative non-schistosoma harboring host because we know that prey-switching and alternative food source can greatly impact the predation pressures that predators can exhibit. We were worried that readers might assume that we were over-estimating predation rates if we forced the predators to only consume snails that can harbor human schistosomes (i.e. by only providing these species). Getting the full suite of potential food sources of snail predators in western Africa was not logistically possible. We felt that adding another snail species was important to offer an alternative prey item and improved the robustness of our findings.

*We have revised the ms to make it clearer why we added the local snail species (lines 82-84), and the possibility that the results could depend on the availability of alternative prey items. We also added to the ms that the effects of the treatments on the local snail species did not significantly differ from the effects on *Bi. glabrata* or *Bu. truncatus*, emphasizing the robustness of our findings (lines 103-106).*

Furthermore, the manuscript is vague and unclear about what “snail densities” are throughout. Does this mean only the potential host snails, all snails or either *Biomphalaria glabrata* or *Bulinus truncatus*? It seems like the authors are lumping all the snails together and it makes it very difficult to separate out the true effects and interpretation.

We apologize for the lack of clarity. We now emphasize that the effects of treatments did not differ among the three snail species (lines 103-106). We also clarify “snail densities” throughout the ms.

A related to this issue, was how the manuscript lumped schistosomes together at some points, focused on *Bi. glabrata* and *S. mansoni* for some experiments and generally ignored the effects on *Bulinus* and *S. haematobium* and any interactions between the species. For example, both snails and parasites are included in the main mesocosm experiment and in the laboratory egg viability experiment, but only cercariae from *S. mansoni* were tested in the toxicity mesocosms (data in Extended data table 4). Specifically, in extended data table 1 – there are statistically significant effects from the structural equation model of the other species of snails *Bu. truncatus* and *H. cubensis* that are not discussed in the paper. In extended data table 2 only *S. mansoni* infected *Bi. glabrata* are analyzed. Extended data table 3 is unclear about whether these results combined *S. mansoni* and *S. haematobium*. Finally, the epidemiological model is of *S. haematobium*, but the rest of the results are primarily *S. mansoni* (Line 416-417). The manuscript seems to jump between the two species and use them almost interchangeably. While these taxa are often discussed together in other literature, in this manuscript it was confusing and potentially misleading. I would encourage the authors to clarify and be more transparent about their results for both species separately and any

assumptions they are making about the responses of one taxa, whether snail or parasite, for the other.

Good points. We have attempted to revise the ms so that it is more transparent in regards to snail and parasite species we are referring to and better justifies why we focused on certain species in our analyses. An important finding that should have been more thoroughly emphasized in the ms is that the effects of the treatments did not differ among the three snail species. Also, we should have emphasized that we focused on Bu. truncatus for the modeling because we parameterized our models using human infection data from Senegal collected during 2011-2013 at a site where S. haematobium was the predominant schistosome species infecting people and Bulinus spp. are the primary intermediate host for S. haematobium in the region (lines 588-590). Hopefully these points are now clear.

Another area for improvement is the interpretation and context for the strong negative effect of the pesticide chlorpyrifos on the snail predators. First, very few predators were actually added to each mesocosm, particularly only 3 crayfish, so that mortality of even one individual would have a huge effect (>33% or >14% mortality with only one death in crayfish and water bugs respectively) and in a ~10 week long experiment reproduction of predators may not have been possible to replace mortality (unlike the rapid snail population growth). If all the predators died that would be expected to have dramatic effects. Furthermore, the effects of predation were inflated by the lack of structure in the mesocosms for prey refugia and lack of alternative prey for predators. Follow up experiments with snails and predators, or a more thorough incorporation of the literature on predation among these species would be helpful in interpreting whether the large effects of predation are representative or exaggerated by weaknesses of the experimental design. Additional experiments would be easier to support because biosafety precautions associated with the parasite exposure may not be necessary unless predation rate differs between infected and uninfected snails.

We struggled picking our predator densities because we wanted to make sure that we were using realistic densities but also realized that realistic densities might provide small sample sizes. We erred on the side of making sure that the densities were ecologically relevant (lines 103-106). Given the enormously strong effects of the estimated environmental concentrations of these chemicals (killing almost all crustacean predators), we doubt that having more predators per tank would have changed the overall effects. In addition, running the experiment long enough to allow predators to reproduce would have resulted in huge variation in the densities of predators in tanks in which they survived, and would have been highly dependent on the sex ratio of survivors, so we chose to avoid the issue of predator reproduction by ending the experiment before crayfish had a chance to reproduce. Keep in mind, however, that although predators will reproduce in nature, their generation time is much longer than the snails and thus recovery will be delayed relative to the prey population.

To address the reviewers concerns regarding the lack of structure, we now include an experiment where macrophytes were manipulated and also varied in densities when present so that there was a gradient of structure (lines 494-580, Extended Data Figure 2, Extended Data Table 8). In this experiment, we show that realistic densities of the snail predators greatly reduced snail populations and prevented recruitment to the adult snail subpopulation regardless of the density of macrophytes.

In general, I also noticed limited context from the previous literature on the role of predators on infected hosts. Instead, the literature used to support the strong top down effects was based on predation on parasite free-living infective stages, which was not what the study was investigating. Predation on infected hosts is a different mechanism than predation on parasites with potentially other effects on the aquatic communities. The authors should support their results with relevant examples from the extensive literature on the role of predation on infected hosts.

Good suggestion. We have added more relevant citations specific to predation of schistosome-infected snails and particularly highlight the work of Swartz et al (2015) which explicitly focuses on predation on infected vs. uninfected Bi. glabrata and Bu. truncatus.

Unfortunately, I believe these issues, as well as other minor issues provided in the specific comments below, prevent the manuscript from being acceptable in the present form. However, I do believe that there is also significant merit in the research and if the authors can make substantial improvements in these areas that a resubmission of the manuscript should be considered.

We thank the reviewer for support of the merit and resubmission potential of our work.

Specific Comments

The manuscript was an appropriate length and written concisely, but there are several important issues that could be clarified specifically related to the experimental design and interpretation of the mesocosm experiment. There were also a couple instances where I thought the authors were overselling their claims and made some suggestions to highlight the importance of the findings without going too far. Finally, there were some instances where more methodological detail is needed. One area is the parasite egg amounts and timing to the mesocosms and how this interacts with snail density. In the methods and extended data section the authors did provide adequate information so that others could replicate mathematical modeling as well as detailed information about the biosafety precautions to the researchers and the native environment to prevent infection or release of organisms. However, for interpretation it would be helpful to include information about the densities of organisms at the end of the experiment, specifically mortality data on the predators, which is currently only incorporated into variables for modeling. Fig1 shows many key relationships, but are actually taken from the path analysis rather than actual experimental values and makes it difficult to compare to previous literature because of the unit-less, experiment specific measures.

These concerns are addressed below. Thank you.

Abstract:

Line 23: Explain what is meant by intensification. Does this mean prevalence, intensity, or some other measure?

We have clarified that both prevalence and infection intensity have increased.

Line 25: Explain increases. Does this mean increase in area irrigated or permanency? The term is unclear.

Good point. We meant to encompass both area irrigated and permanency of standing water, but recognize that this was unclear. To maintain brevity in the abstract, we have avoided the confusion by more broadly stating that water management practices such as dam construction have increased snail habitat.

Line 29: Where is the agrochemical use predicted to increase by 2050? Is this globally or in areas where schistosomes could be present. Please be more specific.

This is now more specific. It is mostly in tropical areas where schistosomiasis is endemic. We have clarified this in lines 167-170, rather than in the abstract to stay within word limits in the abstract.

Line 30: I think this statement extends the results too far. While the combined experiments and modeling demonstrate potential mechanisms that could lead to increasing schistosomiasis generally the applicability is far from global and many other conditions where not considered (for example, prey switching by predators, refugia for prey) that would potentially influence the results across both small and large geographic scales. I recommend revising to better reflect the scope of the results.

We have toned down this claim.

Line 37: Insert “agricultural” after “identifying”.

Done.

Main text:

Line 48: delete “cycle”

Done.

Line 49: Please define the terms “definitive” and “cercariae”. Readers may be unfamiliar with the meaning of these terms.

Done.

Line 54: Are there more up to date values of infection risk? 2003 is quite old.

We have updated these.

Line 70: What is the rationale for including both *Biomphalaria glabrata* and *Bulinus truncatus*? Furthermore, what is the rationale for including the *Haitia cubensis*? How does this represent the natural environment the mesocosms should be modeling? See the general comments above for additional considerations.

*We have now included the rationale for including all these hosts. The rationale for including both *Biomphalaria* and *Bulinus* snails is that they are often syntopic and both are important vectors of human schistosomiasis. The rationale for including *H. cubensis* is that it functioned as an alternative food source so we weren't forcing the predators to only consume snails that*

can harbor human schistosomes. Also, H. cubensis naturally co-occurs with Bi. glabrata throughout much of the Caribbean.

Line 79-80 & Line 248-259: How many eggs were added to each mesocosm for each parasite species? How did the numbers of eggs added on different days compare? Both this section and the methods are vague and do not report the actual numbers added. While it is stated that this would simulate egg introduction from humans in an endemic setting there is no information given about how the density of egg input represents natural communities.

This is provided in the supplement on lines 298-303. We should have referred the reader to the supplement here and now do so.

Furthermore, how did the timing of egg addition influence the results? For example, snails were in the tanks reproducing uninfected for about 3 weeks before parasite addition. I presume that agrochemical induced changes could result in higher parasite densities where parasites were added. However, the parasites castrate the snail, which could influence reproduction, so it is unclear how high infection rates of the snails would then influence snail density. How does the timing of the mesocosm additions represent a natural system?

We added a section to the supplement discussing how the timing of egg and agrochemical additions might have influenced the results (lines 284-287). We also added to the supplement a discussion on how alternative prey resources for the predators could affect the results (lines 540-560, Extended Data Figure 1, Extended Data Table 8). We think these caveats have greatly improved the ms and also help to identify important future research directions.

Line 83: Insert “laboratory” before “toxicity”

Done.

Lines 104-106: Here and elsewhere (Line 351-355), the authors draw from studies on the diversity of predators on parasite life stages to support their results of predation having an important effect on transmission rates. However, the types and effects of the predation are not the same. Predation on parasite life stages could reduce the standing crop of infective stages, but the predation on the intermediate hosts eliminates further parasite production by killing the host. These are two distinct pathways that are not necessarily interchangeable. It is unclear if these predators could consume the parasites themselves in addition to their effects on the snail population. Furthermore, there is extensive literature on the role of predation on disease dynamics and that could be better incorporated here as well as elsewhere (Line 351-355) in the manuscript.

The study we reference included consumption of snail hosts as well as more direct consumption of cercariae. However, we have provided additional citations to support this claim here and elsewhere.

Line 109: Please see the general comments above for additional discussion. The term “snail densities” is used throughout, but it is unclear if the authors are talking about all three snails, just the schistosome host snails (*Bi. glabrata* and *Bu. truncatus*), or one or the other. These results are difficult to interpret without more information about the study design. Throughout the results the authors need to be explicit about what the responses are and by which taxa,

particularly when the results tables are only included in the extended tables making it difficult for the reader to access them directly within the manuscript. In the methods when the analyses are discussed and in the discussion of the results the distinction and effects of the different snails should be made clearly. Results for the non-host snail are presented in Extended table 1 and are reported as being statistically significant, but this is never mentioned in the manuscript.

We now 1) are clearer throughout regarding snail densities, 2) discuss the results on the non-host snail, and 3) highlight that the effects of treatments on the three snail species did not differ significantly (lines 103-106).

Line 118: Remove “indicating that the density of infected snails should reliably indicate schistosome exposure risk to humans”. The study did not directly test this and it is an overstatement of the experimental results.

Done.

Line 141-144: How similar are these species? I would like to see further evidence to support this claim.

This is now provided on lines 172-173, 541-543.

Figures:

Fig. 1A: I think the figure is misleading by not including the pathways from fertilizer and atrazine. The lack of numerical values by the lines makes it look as if these pathways don't have significant effects when they do. The representation of the figure over-emphasizes the role of predation, while the bottom-up factors are pushed back to extended data figure 1. While I can understand the authors' desire to keep the figure simple and interpretable, it would be better to have both parts of the figure included.

This has been revised accordingly.

Figure 1E: I would revise the x-axis label to be Agrochemical status or treatment. It was confusing at first to see absent as a value for agrochemical presence.

Great suggestion. We revised the axis.

Furthermore, as stated in the specific comments above, I think it would be helpful to present figures of the actual data rather than the output of the structural equation model. The unit-less measures are not transparent and prevent the data from being compared with the literature.

To facilitate easily extracting the general patterns and conclusions from these figures, we have kept them as is. However, we have added to the figure legend that the raw data are provided in the supplement, which should facilitate the results being compared with the literature.

Fig. 3D. Missing the letter – also the color scheme did not reproduce well in B&W. I would suggest changing the color scheme so that the trends are still apparent when reproduced in other formats.

We have added the letter to panel D, but unfortunately were unable to find a satisfying alternative color scheme that would print well in black and white or grayscale.

Methods:

Line 218: What was the amount of water?

Now provided.

Line 222: What is the rationale/justification for the different snail densities and particularly for the predators?

Now provided (lines 257-260).

Line 326: Does snail density refer only to the potential host snails? Clarify.

Now clarified.

Line 486: The authors acknowledge some lack of realism in the mesocosm experiment, but could a follow up study be done to examine prey-switching and refuge-seeking by the snails with the predators to help examine the strength of these effects and the potential influence on the results?

This is a great suggestion. We tried to address the prey-switching issue by including three snail prey species in the mesocosms. Given that the effects of treatments and predators on these three species of snails did not differ significantly, we believe we have at least indirectly addressed this issue, although we admit that additional research would be useful. We also more thoroughly address the prey-switching issue in the ms (lines 494-580, Extended Data Figure 2, Extended Data Table 8).

The impacts of macrophyte structure and refuge-seeking behavior of the snails were not previously addressed, so in the revised ms we have added the results of a separate mesocosm experiment examining the effects of macrophyte structure on snail abundances (see lines 494-580). The density of macrophytes did not alter snail predation rates and in both cases, the predators strongly regulated the snail populations. Furthermore, our mathematical model implicitly accounts for prey-switching behavior by the predator by using a Holling type III functional response which allows the per-capita, per-predator predation mortality of snails to become increasingly low with low snail abundance. We have highlighted this on lines 551-555. We also added a more extensive discussion of the potential effects of refugia with reference to existing literature on snail refuge use in response to predation cues and infection status (lines 561-580). We hope that this has addressed the reviewer's legitimate concerns and strengthened the ms.

Author contributions

Not all authors indicated on the title page are included in the author contributions (Missing DJC).

This has been fixed. Thank you.

Extended Data

Extended Data Figure 1. Please see comments for Figure 1.

Revised accordingly.

Extended Data Table 1. Please explain the significant effects of the non-host snail as well as clarify the interactions of the two schistosome hosts with each other.

We have now added this content to the ms.

Extended Data Table 2. This table only represents *Bi. glabrata*. What about *Bu. truncatus*?

*There was a sufficient number of infected *Bi. glabrata* alive at the end of the experiment to conduct this analysis, but not for *Bu. truncatus* (because infected *Bu. truncatus* had lower survival rates, insufficient *S. haematobium* miracidia were added to each tank, or for some other reason).*

Extended Data Table 3. Does this table represent the total infected snail density with *S. mansoni*, *S. haematobium* or just one or the other?

This has been clarified.

Reviewer #3 (Remarks to the Author):

Title: Agrochemical pollution increases risk of human exposure to 2 schistosome parasites

3 Authors: Neal T. Halstead^{1*†}, Christopher M. Hoover², Arathi Arakala³, David J. Civitello⁴, Giulio A. De Leo⁵, Manoj Gambhir³, Steve A. Johnson⁶, Kristin A. Loerns¹, Taegan A. McMahon⁷, Thomas R. Raffel⁸, Justin V. Remais², Susanne H. Sokolow⁵, Jason R. Rohr¹

An interesting and potentially very important topic – proposing that the current trend towards increased agrochemical use can, through increasing snail intermediate host densities (predominantly through reducing snail predators), increase the numbers of infected and uninfected snails, and thereby potentially increase the global burden of schistosomiasis. In particular, using a combination of experimental and modelling simulations, the authors demonstrate that: i) commonly-used agrochemicals can induce mortality in the snail predator population at environmentally relevant concentrations, resulting in R0 estimates equivalent to those in predator-free environments; ii) snail and infected snail densities may also increase due to feeding on increased algae present.

Thank you for the kind words.

The first question raised is whether agrochemical increased usage globally/equally distributed – as schistosomiasis, a disease of animals as well as the humans presented here, is a highly focal disease and impacts the poorest of the poor populations – ones which may not afford agrochemicals on their subsistence farms next to small, often transient, waterbodies? However, the authors do support this with cited evidence in support of mass increases in agrochemical usage within sub Saharan Africa, but may still be worth considering – in particular in relation to the e.g. large lake v small pool waterborne risks of schistosomiasis in Africa.

This is an excellent point. Agrochemical applications are certainly not uniform on relatively small spatial scales. However, agrochemical contamination (and the ecological effects resulting from it) is also not limited only to the sites of application. In countries with higher rates of agrochemical use, such as the United States, agrochemicals and their breakdown products are found in most surface waters even in areas not dominated by agriculture. Although there are certainly other sources of agrochemical inputs (e.g. golf courses, residential lawns, etc.), we feel it is reasonable to expect increases in agrochemical inputs to surface waters in schistosome-endemic regions as usage increases in these countries—even if the chemicals themselves aren't necessarily being used in the poorest communities. Clearly this is a topic that warrants further attention in future research, and we have a recently funded project that launched field research this year to further address these issues in the Senegal study region.

The second, is that how relevant is this to standard snail-predator-free localities, outside those biological-control prawn/crayfish trials successfully ongoing in Senegal and Kenya? However, I do appreciate if snail densities do also increase per se within areas of high agrochemical pollution, relative to pollutant-free sites, then this should be the case.

Some clarification within each of the methodology subsections would be of use to the reader – in regards to what specific hypothesis/aim each component is addressing.

Indeed in general it would have been of interest to know how viable the cercariae were being shed from such infected snails – where one could propose that, whilst potentially increasing the snail densities, could such agrochemicals present in the water be serving as cercariacides and thereby reducing the risk to subsequent human infections? I appreciate no such impact on the eggs was observed.

Great question. We conducted an experiment while completing these revisions that explored whether the three chemicals we tested, at their peak estimated environmental concentrations, caused greater or more rapid cercarial mortality than observed in the controls. The methods and results of this experiment are reported in lines 446-472 and Extended Data Table 5. Importantly, none of the chemicals increased cercarial mortality with up to 12 h of exposure. We hope this satisfies the reviewer's concern and strengthens our ms.

To ensure that schistosomiasis-endemic regions can address their
156 current and pending human malnutrition crises without increasing schistosomiasis, it will be
157 important to implement farming practices that minimize agrochemical runoff

Valid points, wherever feasible. Hence I do think, with some minor clarifications, this article will be of good general interest.

Thank you again for the kind words.

A few specific comments or queries the authors may like to address:

Greater than 10% of the global population is at risk of schistosomiasis,

Interesting way of presenting this, a focal/NTD, but impressive if accurate.

Some potential text proof-reading might help – such as '[The] global human population...' To meet the food demand[s] necessary....

Revised accordingly. Thank you.

Humans act as the
49 definitive host

Humans act as definitive hosts for human schistosomiasis – as of course there are plenty of animal schistosomes also.

Revised accordingly. Thank you.

when cercariae released from snails in infested waters burrow
50 through skin a

Cercariae do penetrate but I'm not sure 'burrow' is the appropriate term.

Replaced "burrow" with "penetrate".

with an
54 estimated 779 million people at risk of infection as of 2003.
55

There are much more recent estimates of numbers at risk since 2003.

Replaced this with a more recent estimate.

common insecticide (chlorpyrifos), a common herbicide (atrazine),

I presume they are two commonly used within schistosome-endemic zones/developing countries. Ok, I see they get to this by lines 76-78 (p4)

They are, as you point out.

(*Biomphalaria glabrata* [the intermediate host of *Schistosoma mansoni*], *Bulinus truncatus* [the 70 intermediate host of *Schistosoma haematobium*],

[an] intermediate host for each – there are many alternatives.

Clarified

Also any details on these snail lines? Presumably laboratory passaged?

Yes. These are snail lines passaged for years by the Schistosomiasis Resource Center. This has been clarified (lines 254-259).

Figure 2. Actual number of infected *Biomphalaria glabrata* as a function of live *Bi. glabrata* at 178 the end of the experiment

Are these real data? How can the number of infected snails go up over time – are the eggs/miracidia not just presented at the beginning of the experiment?

Correct. In the Methods, we highlight that schistosome eggs were added to the tank on three separate occasions (lines 284-287).

Or was it at three time points: Five infected hamsters were euthanatized on 27 July 2010, 4 August 2010, and 12 August 2010),

Yes. They were added to the tank on several occasions

And matched per tank?

Yes. We homogenized the egg solution, made aliquots with similar numbers of viable eggs, and added them to each tank. This is detailed in the methods (lines 292-303).

This has to be made clearer within the text.

We have made all this clearer in the main text.

Methods

Immediately before application

222 of agrochemical treatments on 8 July 2010, snails (27 *Biomphalaria glabrata* (NMRI strain), 11

223 *Bulinus truncatus* (Egyptian strain), and 30 *Haitia cubensis*) and snail predators (3 crayfish 224 (*Procambarus alleni*), and 7 giant water bugs (*Belostoma flumineum*)

Specify why these somewhat random and non-matched numbers of snails were selected?

This was based strictly on availability. We used as many snails as the Schistosomiasis Resource Center could provide that was within ecologically relevant densities (lines 257-260). The same was the case for the non-host snails and predators. Thus, availability and ecologically relevant densities dictated the selected experimental densities.

Mature eggs were

254 stored in a 1.4% NaCl solution to inhibit hatching in a 50 mL centrifuge tube. Eggs were
255 suspended repeatedly using a vortex mixer and sixty-five 3mL aliquots were prepared for
each

256 schistosome species and added to the tanks within two hours of collection.

Ok, but would it not have been better/more standardized to hatch the miradia and give each replicate tank a precise set number?

It indeed would have been more standardized but this would not have been logistically possible for a field mesocosm experiment that included 60 tanks.

Snail infection status was determined by

304 cracking each snail's shell and inspecting the hepatopancreas and gonads under a
dissecting

305 microscope.

Ok, but simply 'shedding' the snails for viable infections at the end of study may have been useful.

*We had >24,000 snails of three species at the end of the study. Sorting out three snail species and living from recently dead snails from 60 1200-L tanks, and then shedding >6,000 *Bi. glabrata* and *Bu. truncatus* (combined) was not logistically possible given our manpower and time constraints in this study. We could have shed a subset, which we did in the "Cercaria Production Experiment" with snails that were infected.*

Cercaria Production Experiment

358 Experimental design

Are the authors now testing for cercarial viability with agrochemical present? Within each subsection of the material and methods, some explanation of the specific hypothesis/aims being tested would be very useful.

Great suggestion. We added hypotheses/aims in each subsection.

Indeed it would be very interesting to know if, whilst potentially increasing the snail densities, could such agrochemicals present in the water be serving as cercariacides and thereby reducing the risk to human infections.

We added the results of a new lab study to address this request (lines 446-472, Extended Data Table 5).

Ah nice, I see they are testing this in terms of the eggs at least, although no impact on egg viability was observed.

Correct!

After applying agrochemicals to each well, approximately 20 eggs of either *S. mansoni* or *S. 403 haematobium* were added to each well.

The modelling used appears appropriate.

Many thanks!

Reviewers' comments:

Reviewer #1 (Remarks to the Author):

Novelty/wider implications: In general, I thank the authors for providing elaborate answers to my comments on the novelty, and their efforts to improve the ms according to the reviewers' suggestions. I do agree with many of their arguments for the novelty of the study, but still (some) of the authors have already shown in a previous study (Sokolow et al 2015, PNAS) and the epidemiological model first developed therein, that a reduction of prawns (predator) will reduce schisto-snail densities and ultimately transmission. Here, they investigate the effect of agrochemicals, expand the existing epidemiological model of 2015, and show that the strongest effect is through a reduction in predators (top-down). And this is indeed a novel finding (for schisto), but is it enough to move the field and a Nature publication? Here I tend to agree with reviewer 2, that even though the study may not move the thinking about the field in a new direction, it certainly reinforces the ideas that agrochemicals can have indirect, negative effects on host-pathogen interactions, and thus identifying and quantifying a potential strong risk factor for schistosomiasis now and in the future. However, to confirm if the results are generalizable across broader geographic areas and habitat types, I still believe the results should be backed by field studies. The author themselves mention that these studies are already planned/begun in Senegal, recognizing the importance of this, and I complement the authors for initiating such important work. Other high impact studies on the snail ecology/population-dynamics and infection risk links, also include field data, for example Perez-Saez et al, PNAS 2016, and Sokolow et al, PNAS, 2015, not to mention the study on the wildlife system (Rohr et al, 2008) that perhaps inspired the authors in the first place. Eventually whether extending the experimental results with an epidemiological mathematical model is enough to address this shortcoming, must of course be an editorial decision..

I would like again under this point to complement the authors for their nice mesocosm set-up, which they now describe in even greater detail. After reading this, I think including one or two pictures of the set-up would be a great addition to the description, if possible.

1: This point is addressed adequately.

2+3: The authors have revised the document to address these points only partly: they discuss prey-switching, but potential effects on the snail community richness; composition (and interactions, i.e. competition with other snail species that also are likely to increase in density) are not touched upon. Given the rather well-established relationship between biodiversity and ecosystem functioning, in schistosomiasis disease ecology sometimes referred to as the decoy effect (see for instance Johnson, P et al, several papers on experimental work and reviews, or Stensgaard et al 2016 for an empirical study), this deserves mentioning. The authors themselves have a paper from 2014 on the effect of agrochemicals on aquatic biodiversity and ecosystem functioning... I recognize adding this level of complexity to an already complex mesocosm system, is probably not feasible, but at least it should be discussed.

4: addressed adequately.

Reviewer #2 (Remarks to the Author):

Halstead et al. have provided a well-written manuscript describing a field mesocosm experiment, follow up mesocosm experiment, small scale laboratory toxicity tests, and epidemiological modeling investigation of the role of agrochemicals on human schistosomes. Combined, this investigation demonstrates several novel insights such as the effects of pesticides on predators of infected hosts in

a mesocosm setting and the incorporation of the data into the epidemiological models.

It is clear from reading the revised manuscript as well as the rebuttal letter several times, that the authors have incorporated many changes and improved the manuscript substantially from the previous version. However, there are still a few remaining issues that I believe need to be addressed prior to a decision on the manuscript. Some of the issues stem from those incompletely addressed from the first reviewers' comments and some stand out based on new information that has been added or clarified in the revised manuscript. Below, I draw from the original reviews and describe these concerns.

Reviewer 2 asked specifically about the community structure in the tanks and the low densities of predators. The authors mention several times that the numbers of predators used were "ecologically relevant" and based on ecological limits, but never cite data or published literature to support densities used. This should be better justified. Is it ecologically relevant in the local area or the schistosomiasis areas the mesocosms are trying to replicate? While it is remarkable that only a few predators are able to have drastic effects on the snail population based on the new experimental information provided, in the main experiment it could be possible that there is a big difference when all three die (100% mortality) than when three out of ten die (30%), even though the same number died, because there would still be some remaining to regulate the snail population. Both of these values might be "ecologically relevant" yet one is much more robust and able to represent the effects of a treatment thought to induce mortality. I wonder if the top down effects are so strong because all the predators died leaving them essentially a no predator treatment. While the effects in nature might be this dramatic, there may be some predators that survive (as acknowledged by the authors in their consideration of the modeling parameters). In this respect, I disagree with the rebuttal, and think that this could have had a change in the overall effect, but need more data from the experiment to evaluate. I would like to see how many tanks had 100% mortality given these very small initial numbers and what the mortality was of predators in non-agrochemical treated tanks and some discussion of this in the manuscript.

Another part of a comment by reviewer 2 asked the authors to explain how the numbers of infected snails could influence reproduction, because snail density is the only significant effect and infected snails (with mature infections) are castrated. This comment was not addressed in the rebuttal letter by the authors.

Again, part of the comment from reviewer 3: was not addressed by the authors:
"The second, is that how relevant is this to standard snail-predator-free localities, outside those biological-control prawn/crayfish trials successfully ongoing in Senegal and Kenya? However, I do appreciate if snail densities do also increase per se within areas of high agrochemical pollution, relative to pollutant-free sites, then this should be the case."

From directly reading the revised manuscript:

Clarify that the significant effects were indirect – they influenced snail density and therefore influenced infected snail density. There were no significant effects directly on infected snails. In nature there are many other factors besides those tested in these mesocosm that could influence snail density. A mesocosm is an excellent way to get at multiple causal pathways, but they are still not the same as field data in capturing reality where these diseases are endemic. It would be easy to state that the effects are mediated by snail density and make it clear that the agrochemicals did not have an effect directly on parasite infected snails.

94: add that temperature was also monitored.

103 – 106: This statement is confusing and misleading. The authors are arguing that the effects they tested significantly increased the densities of infected snails and therefore the potential risk of pathogenic infections to human definitive hosts, but for two of the three snails, including one host snail, the densities were not sufficiently high enough to even analyze. It is unclear what would not be sufficiently high to analyze. The actual densities are not clearly provided in any of the figures, as was suggested by reviewer 2, but instead only the results of the statistical models. The revised figure legend then refers the reader to supplementary materials for the raw data on the snail densities, but is not clear which supplementary material to look at and each table in the extended data table section provides the outcome of a statistical model or parameter values for the modeling. I think it is important that reviewers be able to evaluate the actual densities measured and ultimately for readers of the published manuscript to be able to interpret the actual implications of the results. It might be statistically significant in terms of the effects and not be biologically relevant if so many snails were dead or uninfected at the end of the experiment it doesn't support the claim that there would be an increase of transmission as a result of these treatments. I would hope this would not be the case, but without this information it is impossible to tell. I suggest the authors include figures of the actual population densities of the organisms in the study or report the actual raw data in specific supplementary material (or extended data, here is a good point that terminology needs to be consistent in the manuscript) that is easy to access by the readers. I could not find it in any of the data provided in the manuscript for review. Figure 2 shows the relationship between overall snail density and infected snail density but excludes the mesocosms where no infected *Bi. glabrata* were present. How many replicates were there where none of the snails were infected? How would the results and implications be altered if the replicates with uninfected snails were taken into account? What conditions occurred in the tanks without any infected snails? Extended Data Figure 2 reports the density of snails in the separate mesocosm experiment with macrophytes and predators and as such does not give the numbers in the main experiment.

As stated in the rebuttal letter "There was a sufficient number of infected *Bi. glabrata* alive at the end of the experiment to conduct this analysis, but not for *Bu. truncatus* (because infected *Bu. truncatus* had lower survival rates, insufficient *S. haematobium* miracidia were added to each tank, or for some other reason)." Don't these conditions make the experiment invalid? If the snails and parasites can't survive the experiment then how can you be certain of the effects of what you are testing? At a minimum this requires a more extensive explanation and justification.

106-108 The same reasoning above applies here too with the predator densities. The authors state that the densities were reduced but to what level? Please provide some meaningful values that aren't an abstract unit-less measure from a statistical analysis. Please clarify what is in the supplement vs. the extended data tables so that the actual data can be located.

167-170 This new statement added in the revisions seems too speculative. Human population growth is not the only driver in agrochemical use. This concern is related to a similar comment by reviewer 3 that agrochemical use may not increase in small communities with subsistence farms that would be the most at risk for schistosomiasis. The authors argue that that this is reasonable and that they have a current project to investigate it. I think the current statement is too far and that this should wait until they have the data to actually back it up. There are many other variables involved in agrochemical use besides human population growth.

359 – Were all the 60 tanks used in the analysis or only those that had successfully infected snails? What about those without sufficient infected snails? Clarify.

Check spelling of scientific names throughout. There were errors in places (i.e. 587)

Reviewer #3 (Remarks to the Author):

Referee 3

I am satisfied, with the inclusion now of an apparently additional mesocosm study line, an evaluation of the impact of agrochemicals on (perhaps surprisingly, a lack of impact on) cercarial mortality, and the emphasis on the actual aspects of novelty here over that published before, including by members of this group, that this manuscript has improved sufficiently for publication.

Most importantly, the manuscript certainly benefits from substantial further clarification of the precise methodologies, including sample sizes and proportions, used throughout. As well as providing a more convincing study, this has greatly improved the flow and general clarity of the text.

Reviewers' comments:

Reviewer #1 (Remarks to the Author):

Novelty/wider implications: In general, I thank the authors for providing elaborate answers to my comments on the novelty, and their efforts to improve the ms according to the reviewers' suggestions. I do agree with many of their arguments for the novelty of the study, but still (some) of the authors have already shown in a previous study (Sokolow et al 2015, PNAS) and the epidemiological model first developed therein, that a reduction of prawns (predator) will reduce schisto-snail densities and ultimately transmission. Here, they investigate the effect of agrochemicals, expand the existing epidemiological model of 2015, and show that the strongest effect is through a reduction in predators (top-down). And this is indeed a novel finding (for schisto), but is it enough to move the field and a Nature publication? Here I tend to agree with reviewer 2, that even though the study may not move the thinking about the field in a new direction, it certainly reinforces the ideas that agrochemicals can have indirect, negative effects on host-pathogen interactions, and thus identifying and quantifying a potential strong risk factor for schistosomiasis now and in the future. However, to confirm if the results are generalizable across broader geographic areas and habitat types, I still believe the results should be backed by field studies. The authors themselves mention that these studies are already planned/begun in Senegal, recognizing the importance of this, and I complement the authors for initiating such important work. Other high impact studies on the snail ecology/population-dynamics and infection risk links, also include field data, for example Perez-Saez et al, PNAS 2016, and Sokolow et al, PNAS, 2015, not to mention the study on the wildlife system (Rohr et al, 2008) that perhaps inspired the authors in the first place. Eventually whether extending the experimental results with an epidemiological mathematical model is enough to address this shortcoming, must of course be an editorial decision..

We appreciate Reviewer 1's discussion about the improvements to the first draft of this manuscript, and of the strengths of this paper. We also agree with Reviewer 1 that the inclusion of field data in this paper would be a substantial improvement, but it will be at least two years before the first results of our field work in Senegal will be available. To summarize our previous arguments as to why this manuscript is worthy of publication now, this research is novel because 1) it expands our knowledge of the indirect effects of fertilizers and insecticides on trematode systems that were not addressed in the Rohr et al. 2008 study; 2) it is, to the best of our knowledge, the first manipulative mesocosm experiment on human schistosomes; 3) it links the results of the mesocosm experiments to actual human risk through epidemiological transmission models carefully parameterized on field, laboratory and literature data.

While the inclusion of field data would be ideal, (and we are in the process of scaling up this work), further multi-year delays associated with obtaining and incorporating the results of our upcoming fieldwork would prevent moving the field forward and bringing attention to the potential (and, in our case, documented) effects of agrochemicals on schistosomiasis transmission. We thus believe that the lack of such data should not preclude our current manuscript from publication in Nature Communications.

I would like again under this point to complement the authors for their nice mesocosm set-up, which they now describe in even greater detail. After reading this, I think including one or two pictures of the set-up would be a great addition to the description, if possible.

Thank you for the kind words regarding the experimental set-up. We have included photographs of the mesocosm set-up in the Supplementary Information (Supplementary Fig. 3).

1: This point is addressed adequately.

2+3: The authors have revised the document to address these points only partly: they discuss prey-switching, but potential effects on the snail community richness; composition (and interactions, i.e. competition with other snail species that also are likely to increase in density) are not touched upon. Given the rather well-established relationship between biodiversity and ecosystem functioning, in schistosomiasis disease ecology sometimes referred to as the decoy effect (see for instance Johnson, P et al, several papers on experimental work and reviews, or Stensgaard et al 2016 for an empirical study), this deserves mentioning. The authors themselves have a paper from 2014 on the effect of agrochemicals on aquatic biodiversity and ecosystem functioning... I recognize adding this level of complexity to an already complex mesocosm system, is probably not feasible, but at least it should be discussed.

Thank you for the comments. We had focused on prey-switching behaviors because they were more readily addressed, both in the design of the mesocosm experiments and in the parameterization of the models. However, we agree that we could have been more complete in our discussion of potential effects mediated through the snail community. We have added an additional section to the Supplementary Information (lines 102-118 of the SI) discussing potential community-level effects in open systems with special attention given to how changes in the snail community might affect schistosome transmission.

4: addressed adequately.

Reviewer #2 (Remarks to the Author):

Halstead et al. have provided a well-written manuscript describing a field mesocosm experiment, follow up mesocosm experiment, small scale laboratory toxicity tests, and epidemiological modeling investigation of the role of agrochemicals on human schistosomes. Combined, this investigation demonstrates several novel insights such as the effects of pesticides on predators of infected hosts in a mesocosm setting and the incorporation of the data into the epidemiological models.

It is clear from reading the revised manuscript as well as the rebuttal letter several times, that the authors have incorporated many changes and improved the manuscript substantially from the previous version. However, there are still a few remaining issues that I believe need to be addressed prior to a decision on the manuscript. Some of the issues stem from those incompletely addressed from the first reviewers' comments and some stand out based on new information that has been added or clarified in the revised manuscript. Below, I draw from the original reviews and describe these concerns.

Reviewer 2 asked specifically about the community structure in the tanks and the low densities of predators. The authors mention several times that the numbers of predators used were "ecologically relevant" and based on ecological limits, but never cite data or published literature to support densities used. This should be better justified. Is it ecologically relevant in the local area or the schistosomiasis areas the mesocosms are trying to replicate?

*We have provided better justification of our chosen predator densities with citations from the literature. Densities of *P. alleni* in South Florida exhibit a high degree of spatiotemporal variability, and can exceed 10 crayfish m^{-2} , but are typically significantly lower (Dorn and Trexler 2007 Freshwater Biology). We chose a density on the lower end of observed natural densities (0.6 crayfish m^{-2}) because 1) our initial stocking densities for snail species were also relatively low, 2) crayfish presence/absence (rather than density) determined the variation in snail survival and recruitment in previous mesocosm studies (Halstead et al 2014 Ecology Letters, Dorn and Hafsadi 2015), and 3) we wanted to limit our impact on local source populations of crayfish (given that we had 60 mesocosms to supply with predators). Finally, our initial crayfish densities were nearly identical to the maximum natural prawn densities reported in the literature (~ 0.6 prawns m^{-2} ; Covich et al. 2006 J. N. Am. Benthol. Soc.), which is also the estimated*

density of prawns necessary to extirpate local snail populations and well above the threshold density of prawns needed to reduce/eliminate schistosomiasis locally (~0.3 prawns m⁻²; Sokolow et al 2015 PNAS), but significantly lower than aquaculture densities that can exceed well over 10 prawns m⁻²; Cohen et al. 1983 Aquaculture, Son et al. 2005 Aquaculture Research).

The realized crayfish density in the model (i.e. the density at equilibrium given the other model parameters) was approximately 0.4 crayfish m⁻². These results agreed with the thresholds and general behavior of the Sokolow et al. 2015 PNAS paper and generated reasonable predator densities.

While it is remarkable that only a few predators are able to have drastic effects on the snail population based on the new experimental information provided, in the main experiment it could be possible that there is a big difference when all three die (100% mortality) than when three out of ten die (30%), even though the same number died, because there were still some remaining to regulate the snail population. Both of these values might be “ecologically relevant” yet one is much more robust and able to represent the effects of a treatment thought to induce mortality. I wonder if the top down effects are so strong because all the predators died leaving them essentially a no predator treatment. While the effects in nature might be this dramatic, there may be some predators that survive (as acknowledged by the authors in their consideration of the modeling parameters). In this respect, I disagree with the rebuttal, and think that this could have had a change in the overall effect, but need more data from the experiment to evaluate. I would like to see how many tanks had 100% mortality given these very small initial numbers and what the mortality was of predators in non-agrochemical treated tanks and some discussion of this in the manuscript.

We agree with Reviewer 2's point that having more crayfish per tank would have provided us with more robust results, but for the reasons described above, we chose to use densities on the lower end of the ranges observed for P. alleni in natural conditions. In previous mesocosm experiments (Halstead et al. 2014 Ecology Letters), survival of P. alleni was high in non-insecticide treatments, and negligible in insecticide treatments. In the current mesocosm study, mortality was nearly 100% in all tanks receiving chlorpyrifos. Only one tank receiving chlorpyrifos had any surviving crayfish (2 individuals) out of 25 tanks that received the insecticide. This likely would have been the case even had we used higher densities of crayfish in each tank because the target concentration of chlorpyrifos used in the mesocosm experiment (64 µg/L) was more than twice the observed LC50 value (29.3 µg/L; Halstead et al. 2015 Chemosphere) calculated from the same local population of crayfish.

The reviewer does make an excellent point that our results should be easier for the reader to access and interpret. Therefore, we have added figures of crayfish mortality using the raw data as units of measure to the Supplementary Information (Supplementary Fig. 2). We have also added acknowledgment that field experiments in open systems are needed to examine rates of predator recovery/recruitment following insecticide exposure (lines 103-107 of the SI).

Another part of a comment by reviewer 2 asked the authors to explain how the numbers of infected snails could influence reproduction, because snail density is the only significant effect and infected snails (with mature infections) are castrated. This comment was not addressed in the rebuttal letter by the authors.

We apologize for not addressing this portion of the comment in the initial response. Certainly, infection does affect snail fecundity. However, it can be somewhat troublesome to neatly quantify in a model. The model in this paper differs from that of the Sokolow et al. 2015 PNAS model. Our model made the assumption that Exposed, but not Infected snails reproduce, but we used a slightly lower fecundity rate than in the PNAS model (0.1 snails/snail/day as opposed to 0.16). Still, our model is largely in agreement with the PNAS model in terms of net snail reproduction. Furthermore, our analyses are based on disease-free equilibrium conditions, and therefore these effects would not change any of the R₀-based results as shown in Figure 3. Finally, the exposed/infected snails generally make up too small of a proportion of the population for their reproductive output (or lack thereof) to likely have a meaningful effect on transmission.

Again, part of the comment from reviewer 3: was not addressed by the authors:

“The second, is that how relevant is this to standard snail-predator-free localities, outside those biological-control prawn/crayfish trials successfully ongoing in Senegal and Kenya? However, I do appreciate if snail densities do also increase per se within areas of high agrochemical pollution, relative to pollutant-free sites, then this should be the case.”

The significant, positive bottom-up effects of atrazine and fertilizer (and the interaction between the two) from our mesocosm experiment suggests that, even in the absence of snail predators, these agrochemicals can increase snail densities (and therefore likely increase the density of infected snails as well). This is consistent with previous research, including data on trematode infections in wild populations (e.g., Rohr et al. 2008). However, we have included the following statement in the Discussion to highlight this result (lines 186-189):

“Additionally, our results suggest that applications of the common herbicide atrazine and fertilizer can increase the risk of human schistosomiasis in situations where snail predators, such as prawns, exist at densities too low to effectively regulate snail populations (e.g., where dams have been constructed or in surface irrigation canals).”

From directly reading the revised manuscript:

Clarify that the significant effects were indirect – they influenced snail density and therefore influenced infected snail density. There were no significant effects directly on infected snails. In nature there are many other factors besides those tested in these mesocosm that could influence snail density. A mesocosm is an excellent way to get at multiple causal pathways, but they are still not the same as field data in capturing reality where these diseases are endemic. It would be easy to state that the effects are mediated by snail density and make it clear that the agrochemicals did not have an effect directly on parasite infected snails.

We apologize for not making this thoroughly clear in the initial revision, we have now edited the ms to make this point clearer, as suggested by the reviewer (lines 101, 107-112, 118-119, 130-141).

94: add that temperature was also monitored.

Done.

103 – 106: This statement is confusing and misleading. The authors are arguing that the effects they tested significantly increased the densities of infected snails and therefore the potential risk of pathogenic infections to human definitive hosts, but for two of the three snails, including one host snail, the densities were not sufficiently high enough to even analyze. It is unclear what would not be sufficiently high to analyze. The actual densities are not clearly provided in any of the figures, as was suggested by reviewer 2, but instead only the results of the statistical models. The revised figure legend then refers the reader to supplementary materials for the raw data on the snail densities, but is not clear which supplementary material to look at and each table in the extended data table section provides the outcome of a statistical model or parameter values for the modeling. I think it is important that reviewers be able to evaluate the actual densities measured and ultimately for readers of the published manuscript to be able to interpret the actual implications of the results. It might be statistically significant in terms of the effects and not be biologically relevant if so many snails were dead or uninfected at the end of the experiment it doesn't support the claim that there would be an increase of transmission as a result of these treatments. I would hope this would not be the case, but without this information it is impossible to tell. I suggest the authors include figures of the actual population densities of the organisms in the study or report the actual raw data in specific supplementary material (or extended data, here is a good point that terminology needs to

be consistent in the manuscript) that is easy to access by the readers. I could not find it in any of the data provided in the manuscript for review. Figure 2 shows the relationship between overall snail density and infected snail density but excludes the mesocosms where no infected *Bi. glabrata* were present. How many replicates were there where none of the snails were infected? How would the results and implications be altered if the replicates with uninfected snails were taken into account? What conditions occurred in the tanks without any infected snails? Extended Data Figure 2 reports the density of snails in the separate mesocosm experiment with macrophytes and predators and as such does not give the numbers in the main experiment.

We again apologize for the lack of clarity in these results and accessing the raw data, and have revised the text and supplementary information to make these results more accessible.

*We have reworded the original statement in lines 103-106 (now lines 109-113) to make it clearer that these limitations only applied to the analysis of infected snails, but that reproductive output and final densities were sufficiently high for all three species to determine that the indirect effects of agrochemical treatments were significant and consistent. The statement now reads as follows (addition in bold): “While *Bi. glabrata* was the only snail species for which a sufficient number of infected individuals were alive at the conclusion of the experiment for analysis **of the relationship between final snail density and infected snail density**, treatment effects on the reproductive output and final densities of all three snail species were significant and in the same direction (Fig. 1A; Supplementary Fig. 1; Supplementary Table 1).”*

Regarding Figure 2, we have made it clearer that while the figure is only showing mesocosms in which infected snails were present, this is for visualization purposes only. The statistical analysis was performed on all tanks using a zero-inflated model. However, because there are two separate pieces of such a model (a binary infected or not portion and a count portion given infection), we chose to display the count portion of the model to make the visual interpretation clearer.

As stated in the rebuttal letter “There was a sufficient number of infected *Bi. glabrata* alive at the end of the experiment to conduct this analysis, but not for *Bu. truncatus* (because infected *Bu. truncatus* had lower survival rates, insufficient *S. haematobium* miracidia were added to each tank, or for some other reason).” Don’t these conditions make the experiment invalid? If the snails and parasites can’t survive the experiment then how can you be certain of the effects of what you are testing? At a minimum this requires a more extensive explanation and justification.

*Again, we apologize for any confusion here. There are a number of reasons for why we might have had lower infection rates for *Bu. truncatus* than *Bi. glabrata* in our mesocosm experiment, which we listed previously and repeat below, that might explain why we did not have sufficient densities of infected *Bu. truncatus* at the end of the experiment to conduct the same analysis as we did for *Bi. glabrata*. Reasons for lower infection rates for *Bu. truncatus* in the mesocosm experiment could include, for example, adding an insufficient number of miracidia to each tank to achieve sufficiently high prevalence, suboptimal size structure of the *Bu. truncatus* populations in mesocosms at the times of *S. haematobium* egg additions (resulting in fewer infected individuals), increased mortality rates of infected *Bu. truncatus* (combined with non-constant egg additions), among other potential scenarios. However, none of these reasons invalidates the observed results on *Bu. truncatus* reproductive output and final densities, which are consistent with both *Bi. glabrata* and the local snail species (which couldn’t be infected at all). It is important to highlight again that snail reproductive output and final densities were strongly positively correlated for all three snail species. Only the prevalence measures from *Bu. truncatus* were not interpretable, and thus dropped from final analysis.*

The most notable drawback of our experimental design is that we were unable to follow the population of infected snails through time during our mesocosm experiment. We had initially hoped to do so, but unfortunately logistical constraints made this impractical. The primary roadblock to following infection through time is that it wasn’t logistically possible to screen live snails for shedding cercariae throughout

the experiment, given the sometimes very high densities of live snails in the tanks and the increased risk of researchers being exposed to agrochemicals and infectious cercariae. Thus, our data on infected snail densities reflects only a snapshot of time in which we anticipated the infected snail densities to best reflect the effects of exposure to miracidia during the period of schistosome egg additions.

106-108 The same reasoning above applies here too with the predator densities. The authors state that the densities were reduced but to what level? Please provide some meaningful values that aren't an abstract unit-less measure from a statistical analysis. Please clarify what is in the supplement vs. the extended data tables so that the actual data can be located.

We have reformatted the manuscript to comply with the style guidelines of Nature Communications and make it clearer where the raw data are available and in what units.

167-170 This new statement added in the revisions seems too speculative. Human population growth is not the only driver in agrochemical use. This concern is related to a similar comment by reviewer 3 that agrochemical use may not increase in small communities with subsistence farms that would be the most at risk for schistosomiasis. The authors argue that that this is reasonable and that they have a current project to investigate it. I think the current statement is too far and that this should wait until they have the data to actually back it up. There are many other variables involved in agrochemical use besides human population growth.

We have modified this section to more clearly state that there is a great deal of uncertainty regarding the distribution of future agrochemical use in schistosome-endemic regions.

359 – Were all the 60 tanks used in the analysis or only those that had successfully infected snails? What about those without sufficient infected snails? Clarify.

We have clarified the sample size used in the analysis.

Check spelling of scientific names throughout. There were errors in places (i.e. 587)

We have corrected these errors.

Reviewer #3 (Remarks to the Author):

Referee 3

I am satisfied, with the inclusion now of an apparently additional mesocosm study line, an evaluation of the impact of agrochemicals on (perhaps surprisingly, a lack of impact on) cercarial mortality, and the emphasis on the actual aspects of novelty here over that published before, including by members of this group, that this manuscript has improved sufficiently for publication.

Most importantly, the manuscript certainly benefits from substantial further clarification of the precise methodologies, including sample sizes and proportions, used throughout. As well as providing a more convincing study, this has greatly improved the flow and general clarity of the text.

Many thanks!

REVIEWERS' COMMENTS:

Reviewer #2 (Remarks to the Author):

Halstead et al. have taken great care to address the remaining reviewers comments in the most recent rebuttal letter. The main points of the manuscript are now more clearly articulated. I particularly appreciate the greater inclusion of the raw data in Supplementary Figure 1 and 2. The most important contributions to this draft are the improvements in clarity of the analysis indicating that the effects were indirect in that the tested variables acted on snail densities, which were then related to infected *Bi. glabrata*. In my mind there are still issues with the experiment when trying to extend the data on the link between density and potential infection of the other schistosome host *Bu. truncatus*, but I believe this current version is presented in such a way that the readers can interpret it for themselves.

I have two final, minor, questions/concerns. The first is that the authors provide a very detailed explanation of the rationale for the density of the crayfish and the effect of chlorpyrifos in the rebuttal letter, but any change to the manuscript itself was pretty non-existent. I believe that this rationale is important in understanding the effects of the experiment along with the knowledge that the mortality was nearly 100%. I agree with the explanation of this information provided in the rebuttal letter, but would expect to see this provided for in the text or at least in the supplement.

Second, why was this high of a pesticide value used if it was known that it was likely to induce 100% mortality? The information about the LC50 value should be included in the methods of the study or at least in the discussion potentially around lines 183-186. This is apparently not a new result of this study and it should not present as if it is.

I believe that these additional points will help the reader understand the specific context of the experiment and judge whether or not this situation is likely to represent natural conditions as well as where the study sits in terms of previous research. I believe these should be straightforward changes as the text in the rebuttal letter will be a good starting place.

REVIEWERS' COMMENTS:

Reviewer #2 (Remarks to the Author):

Halstead et al. have taken great care to address the remaining reviewers comments in the most recent rebuttal letter. The main points of the manuscript are now more clearly articulated. I particularly appreciate the greater inclusion of the raw data in Supplementary Figure 1 and 2. The most important contributions to this draft are the improvements in clarity of the analysis indicating that the effects were indirect in that the tested variables acted on snail densities, which were then related to infected *Bi. glabrata*. In my mind there are still issues with the experiment when trying to extend the data on the link between density and potential infection of the other schistosome host *Bu. truncatus*, but I believe this current version is presented in such a way that the readers can interpret it for themselves.

Thank you for the kind words.

I have two final, minor, questions/concerns. The first is that the authors provide a very detailed explanation of the rationale for the density of the crayfish and the effect of chlorpyrifos in the rebuttal letter, but any change to the manuscript itself was pretty non-existent. I believe that this rationale is important in understanding the effects of the experiment along with the knowledge that the mortality was nearly 100%. I agree with the explanation of this information provided in the rebuttal letter, but would expect to see this provided for in the text or at least in the supplement.

We have added this information to the methods section (lines 224-235).

Second, why was this high of a pesticide value used if it was known that it was likely to induce 100% mortality? The information about the LC50 value should be included in the methods of the study or at least in the discussion potentially around lines 183-186. This is apparently not a new result of this study and it should not present as if it is.

We chose the target concentrations of agrochemicals for this experiment in a non-arbitrary way by calculating the estimated peak environmental concentration (EEC) using the U.S. EPA's GENEEC (v2.0) software (see lines 240-244, Supplementary Table 7). This software uses the physicochemical properties of the agrochemical and manufacturer-recommended application rates from the product label to generate an estimate of the average peak environmental concentration in surface water using simulations from a standardized scenario. While this method ignores any information about toxicity to target or non-target organisms, we felt that this was the most defensible approach to selecting agrochemical concentrations that would be considered environmentally relevant. We have added an emphasis on the environmental relevance of the selected agrochemical concentrations to the Methods on lines 246-247.

We do acknowledge that the high toxicity of chlorpyrifos to crayfish at environmentally relevant concentrations is not a new result of this study and have modified the discussion to refer to published literature (line 193-196).

I believe that these additional points will help the reader understand the specific context of the experiment and judge whether or not this situation is likely to represent natural conditions as well as where the study sits in terms of previous research. I believe these should be straightforward changes as the text in the rebuttal letter will be a good starting place.

We agree, and thank the reviewer for their suggestions.